# BIDIRECTIONAL GENERATIVE RETRIEVAL WITH MULTI-MODAL LLMS FOR TEXT-VIDEO RETRIEVAL

## ABSTRACT

In recent years, multi-modal large language models (MLLMs) have shown outstanding advancement in various multi-modal understanding tasks by leveraging the powerful knowledge of large language models (LLMs). Extending MLLMs to text-video retrieval enables handling more complex queries with multiple modalities beyond simple uni-modal queries for traditional search engines. It also provides a new opportunity to incorporate search into a unified conversational system, but MLLM-based text-video retrieval has been less explored in the literature. To this end, we investigate MLLMs' capabilities in text-video retrieval as a generation task, namely, generative retrieval, in two directions. An intuitive direction is *content generation* that directly generates the content given a query. Another direction is *query generation*, which generates the query given the content. Interestingly, we observe that in both text-to-video and video-to-text retrieval tasks, query-generation less suffers from the bias and significantly outperforms content-generation. In this paper, we propose a novel framework, Bidirectional Text-Video Generative Retrieval (BGR), that handles both text-to-video and video-to-text retrieval tasks by measuring the relevance using two generation directions. Our framework trains MLLMs by simultaneously optimizing two objectives, *i.e.*, video-grounded text generation (VTG) and text-grounded video feature generation (TVG). At inference, our framework ensembles predictions by both generation directions. We also introduce a Prior Normalization, a simple plug-and-play module, to further alleviate the *prior bias* induced by the likelihood of uni-modal content data that often overwhelms the relevance between query and content. Our extensive experiments on multi-modal benchmarks demonstrate that BGR and Prior Normalization are effective in alleviating the prior bias, especially the text prior bias from LLMs' pretrained knowledge in MLLMs, achieving state-of-the-art performance.

## 1 INTRODUCTION

In recent years, large language models (LLMs) (Chowdhery et al., 2023; Touvron et al., 2023; Achiam et al., 2023; Jiang et al., 2023; Chung et al., 2022; Team, 2023) have gained attention due to their powerful knowledge obtained during pretraining. To leverage LLMs' knowledge in understanding visual recognition, LLMs have been extended into Multi-modal large language models (MLLMs) (Alayrac et al., 2022; Driess et al., 2023; Liu et al., 2023b; Li et al., 2024b; Zhu et al., 2023; Chen et al., 2024c;b; Liu et al., 2024) and they have demonstrated remarkable capabilities in comprehensive image and video understanding. Despite the success of MLLMs in various multi-modal understanding tasks, adapting those models to text-video retrieval (*i.e.*, text-to-video and video-to-text), which aims to find the most relevant video (or text) *content* among content candidates given a text (or video) *query* as in Fig. 1a, has been less explored in the literature.

In general, MLLMs are trained to generate a text sequence given the image or video, enabling them to follow complex human instructions regarding visual information. Therefore, integrating the retrieval task into MLLMs can be easily extended to instruction-based retrieval beyond standard retrieval benchmarks with simple captions, in real-world chatting systems. Specifically, MLLMs can be applied to process more complex multi-modal queries to retrieve the most relevant multi-modal content as in Fig. 1b. The versatility, in addition to MLLMs' powerful knowledge, motivates our work to integrate the retrieval task into MLLMs by mainly exploring the Text-Video Retrieval task.

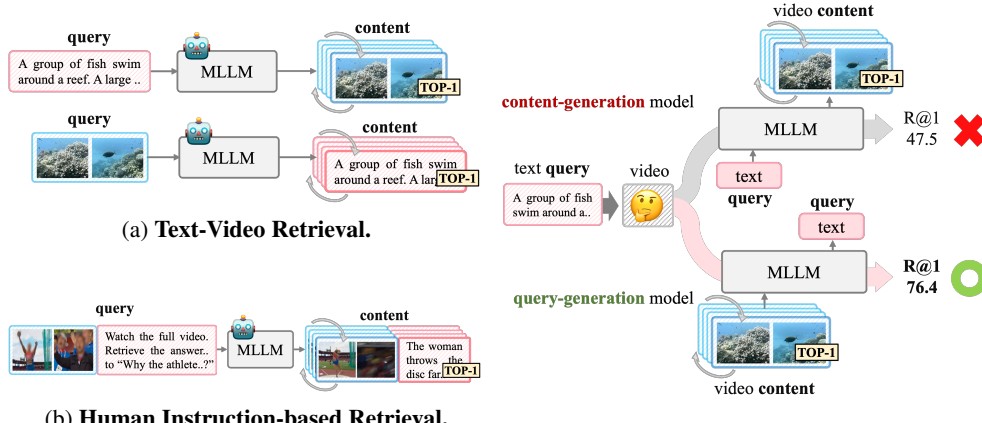

Figure 1: (a) illustrates the standard Text-Video Retrieval task. (b) illustrates the retrieval task based on human instruction with a multi-modal query asking to retrieve the multi-modal content. (c) shows that the query-generation model significantly outperforms the content-generation model.

In this paper, we formulate the Text-Video Retrieval task as a generation task by directly leveraging MLLMs' generation capabilities, *i.e.*, generative retrieval (Tay et al., 2022). Intuitively, when adapting the generative retrieval to Text-Video Retrieval, one might want to generate a content modality given a query modality, *i.e.*, *content-generation*. However, the *query-generation* model, which generates the query given the content, significantly outperforms the content-generation model. For example, in text-to-video retrieval (T2V), the query-generation model which generates the text query given the video content surpasses the content-generation model which generates the video content by a margin of 28.9 on R@1 in Fig. 1c. This is attributed to an undesirable bias towards the prior probability of the content, *i.e.*, *prior bias*. Specifically, on video-to-text retrieval (V2T), the content-generation model generating text content predominantly retrieves the caption with the highest text prior probability in Fig. 2a, *i.e.*, RANK-1 based on the text prior probability $P(\mathbf{t})$). In fact, this caption is retrieved for approximately 777 out of 1,003 videos, accounting for 77% of the total. This implies that MLLMs provide suboptimal retrieval results in V2T ignoring the relevance between the video query and the text content. We also observe that the text prior bias in V2T is more severe than the video prior bias in T2V due to the LLMs' strong pretrained knowledge. Such text prior bias is prevalent even in other multi-modal tasks like visual question answering and captioning (Niu et al., 2021; Ramakrishnan et al., 2018; Cadene et al., 2019; Leng et al., 2024). Overall, both text prior bias and video prior bias are exhibited in MLLMs with the content-generation model for generative retrieval.

To handle both video and text retrievals, we propose a novel framework, Bidirectional Text-Video Generative Retrieval (BGR), which measures the relevance between the video and text using two generation directions. Our framework trains MLLMs by simultaneously optimizing two objectives, *i.e.*, video-grounded text generation (VTG) and text-grounded video feature generation (TVG). At inference, BGR ensembles predictions by both generation directions. For instance, in text-to-video retrieval, BGR bolsters its predictions from content generation (video) by query generation (text), which is more accurate and robust to prior bias as in Fig. 1c. We also propose a plug-and-play module, Prior Normalization, which is a simple yet effective training-free score calibration method to further reduce the prior bias at retrieval inference. We showcase the effectiveness of our BGR equipped with Prior Normalization on four popular Text-Video Retrieval benchmark datasets (DiDeMo (Anne Hendricks et al., 2017), ActivityNet (Caba Heilbron et al., 2015), LSMDC (Rohrbach et al., 2017), and MSRVTT (Xu et al., 2016)). Our in-depth analyses showcase that Prior Normalization can be extended to a Prior Normalized Decoding for a wide range of multi-modal understanding tasks beyond retrieval.

To sum up, our **contributions** are as follows:

- We formulate text-video retrieval as a generation problem by MLLMs. To the best of our knowledge, this paper is the first work that studies *prior bias* induced by the likelihood of uni-modal content data, which is severer in content generation than query generation.

- We propose a novel retrieval framework, Bidirectional Text-Video Generative Retrieval (BGR), to leverage MLLMs' generative capabilities, which is composed of two objectives: video-grounded text generation (VTG) and text-grounded video feature generation (TVG).

- We also present simple yet effective score calibration methods, Prior Normalization and Prior Normalized Decoding, which further reduce the prior bias, leading to performance improvements in BGR as well as in various MLLMs.

- Our extensive analyses demonstrate that the query-generation model is especially effective in mitigating the text prior bias arising from strong LLMs' knowledge in MLLMs, and Prior Normalized Decoding successfully reduces the reliance on the textual data and encourages the model to balance its consideration between the visual and the text modality in various multi-modal understanding tasks.

## 2 PRELIMINARY AND MOTIVATION

### 2.1 PRELIMINARY

Text-Video Retrieval consists of two sub-tasks, *i.e.*, text-to-video retrieval (T2V) and video-to-text retrieval (V2T), aiming to find the most relevant video (or text) content given a text (or video) query. Previous approaches have conducted the retrieval task by employing video-text contrastive (VTC) and video-text matching (VTM) loss functions to align video and text features. The **VTC** loss focuses on aligning uni-modal video and text representation by maximizing the mutual information with the symmetric contrastive loss (Bain et al., 2021). Given the global features of video and text from each uni-modal encoder, $\tilde{v} \in \mathbb{R}^D$ and $\tilde{t} \in \mathbb{R}^D$, the VTC loss is defined as:

$$\mathcal{L}_{\text{vtc}} = -\frac{1}{B} \sum_{n=1}^{B} \left( \log \frac{\exp\left(\tilde{v}^{(n)\top} \tilde{t}^{(n)}\right)}{\sum_{m=1}^{B} \exp\left(\tilde{v}^{(n)\top} \tilde{t}^{(m)}\right)} + \log \frac{\exp\left(\tilde{v}^{(n)\top} \tilde{t}^{(n)}\right)}{\sum_{m=1}^{B} \exp\left(\tilde{v}^{(m)\top} \tilde{t}^{(n)}\right)} \right), \quad (1)$$

where $B$ is the batch size, and $\tilde{v}^{(n)}$ and $\tilde{t}^{(n)}$ denote the global features of the $n$-th video and text. On the other hand, the **VTM** loss predicts whether the given pair of video and text is a match (positive) or an unmatch (negative) pair, *i.e.*, a binary classification task, through a cross-modal fusion score. One special token $z$ is appended at the end (or start) of the sequence to aggregate information from the previous (or succeeding) tokens including both video and text in the cross-modal encoder. Finally, the output feature of the special token, $\tilde{z}$, is used in the binary classification. In inference, the VTM scores generally demonstrate superior performance compared to VTC scores but require larger computational costs. Hence, in the literature (Li et al., 2023b;d), a hybrid approach has been explored, which efficiently retrieves the top-$k$ candidates by VTC, and then computes more accurate scores by VTM for all pairs of the top-$k$ candidates.

### 2.2 MOTIVATION

To analyze the phenomenon where the query-generation model outperforms the content-generation model that suffers from the prior bias, we revisit the inference procedure of T2V, which is as follows:

$$\mathbf{v}^* = \arg\max_{\mathbf{v}} P(\mathbf{v}|\mathbf{t}_{\text{query}}) = \arg\max_{\mathbf{v}} \frac{P(\mathbf{t}_{\text{query}}|\mathbf{v})P(\mathbf{v})}{P(\mathbf{t}_{\text{query}})} = \arg\max_{\mathbf{v}} P(\mathbf{t}_{\text{query}}|\mathbf{v})P(\mathbf{v}). \quad (2)$$

In Eq. (2), to search for the most relevant video content $\mathbf{v}^*$ given a text query $\mathbf{t}_{\text{query}}$, the model takes into account the prior probability of video content $P(\mathbf{v})$ as well as the conditional probability $P(\mathbf{t}_{\text{query}}|\mathbf{v})$ which measures the relevance between text query and video content. Similarly, the inference procedure of V2T is $\mathbf{t}^* = \arg\max_{\mathbf{t}} P(\mathbf{v}_{\text{query}}|\mathbf{t})P(\mathbf{t})$, where the text prior probability $P(\mathbf{t})$ is reflected to find an optimal text content given video query. We observe that when predicting V2T, the model tends to retrieve the caption, which is less related to the video query $\mathbf{v}_{\text{query}}$ but has a high text prior probability (and vice versa), *i.e.*, the prior probability of text content overwhelms the relevance between text content and video query. Fig. 2b shows a representative example. The 'Caption 2' is incorrectly retrieved by two irrelevant video queries since its text prior probability (*i.e.*, uni-modal likelihood $P(\mathbf{t}) = 0.38$) is much higher than the other captions ($P(\mathbf{t}) = 0.00$). Hence, the posterior probability $P(\mathbf{t}|\mathbf{v})$ of 'Caption 2' is significantly higher than the others, leading to

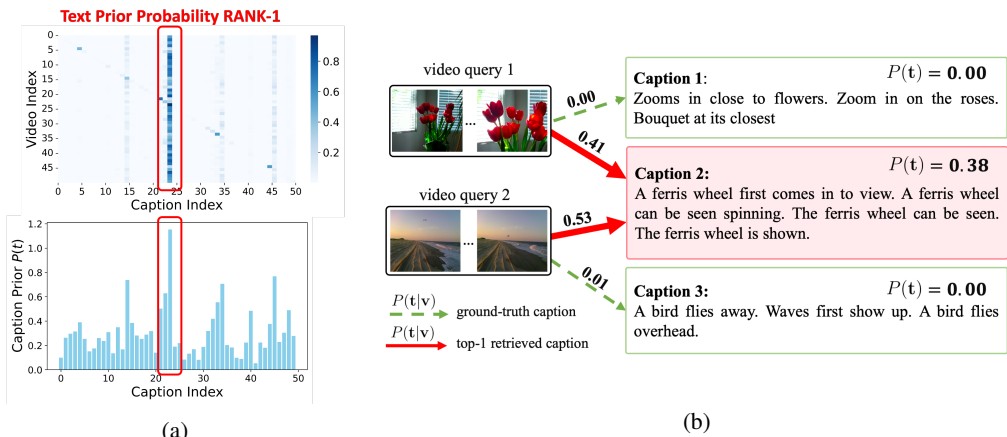

(a)                                                         (b)

Figure 2: **Result of MLLM's prediction on Video-to-Text Retrieval.** (a) The upper heatmap illustrates the retrieval result on the content-generation model in V2T. 50 text-video pairs are sampled to avoid visual clutter. The lower bar plot shows the prior probability of each caption. We observe that the caption with the highest prior probability is retrieved by most videos. (b) illustrates an example of MLLM's prediction on V2T.

the suboptimal retrieval result. This is attributed to the powerful knowledge of LLMs that MLLMs excessively rely on rather than considering the visual contents, *i.e.*, the model exhibits the 'text prior bias'. This text prior bias is commonly observed in various multi-modal tasks like visual question answering and captioning as well as the retrieval task (Niu et al., 2021; Ramakrishnan et al., 2018; Cadene et al., 2019; Leng et al., 2024). Similarly, MLLMs also exhibit the 'video prior bias' on T2V, resulting in the significant performance gap between the content-generation and query-generation model as in Fig. 1c. Further discussion on the video prior bias is provided in Sec. E.1. As a result, we argue that alleviating the prior bias is crucial when MLLMs are used in Text-Video Retrieval.

## 3 METHOD

In this section, we first introduce a simple non-generative baseline for MLLM-based Text-Video Retrieval that utilizes VTC and VTM losses in Sec. 3.1. Then, we present a novel framework, Bidirectional Text-Video Generative Retrieval (BGR) in Sec. 3.2. BGR trains MLLMs with two objectives: video-grounded text generation (VTG) and text-grounded video feature generation (TVG). During inference, BGR ensembles predictions by both generation directions. Lastly, we present a plug-and-play score calibration method, Prior Normalization, which reduces the influence of priors even in the content-generation model, and we extend it for various multi-modal tasks by introducing Prior Normalized Decoding in Sec. 3.3.

### 3.1 MLLM BASELINE FOR TEXT-VIDEO RETRIEVAL

We here introduce a simple baseline of MLLM for Text-Video Retrieval by applying common retrieval objectives, VTC and VTM. MLLMs are generally composed of a Vision Encoder, Projection Layer, and LLM. The visual tokens are first extracted from the vision encoder and then projected into the LLM's embedding space via the projection layer. Those tokens are concatenated to the text tokens and fed to the LLM to perform multi-modal tasks. We fine-tune pretrained VideoChat2 (Li et al., 2024b) which adopts the Q-Former and a linear layer as the projection layer and LoRA to fine-tune an LLM, as illustrated in Fig. 3a. In addition to the projected $L_v$ video tokens $\mathbf{v} = [v_1, \ldots, v_{L_v}] \in \mathbb{R}^{L_v \times D}$ and $L_t$ text tokens $\mathbf{t} = [t_1, \ldots, t_{L_t}] \in \mathbb{R}^{L_t \times D}$ with the hidden dimension $D$, we append two additional special tokens $v_{\text{vtc}} \in \mathbb{R}^D$ and $t_{\text{vtc}} \in \mathbb{R}^D$ that indicate the end of the video and text, respectively. These special tokens aggregate information of each modality and their output representations ($\tilde{v}_{\text{vtc}}$ and $\tilde{t}_{\text{vtc}}$) are used to calculate the VTC loss in Eq. (1). The cross-modal attention is masked to prevent information leakage from the other modality when obtaining $\tilde{v}_{\text{vtc}}$ and $\tilde{t}_{\text{vtc}}$. Similarly, one special token $z_{\text{vtm}}$ is appended to calculate the VTM loss. We update the parameters only in the linear projection layer and LoRA for parameter-efficient fine-tuning.

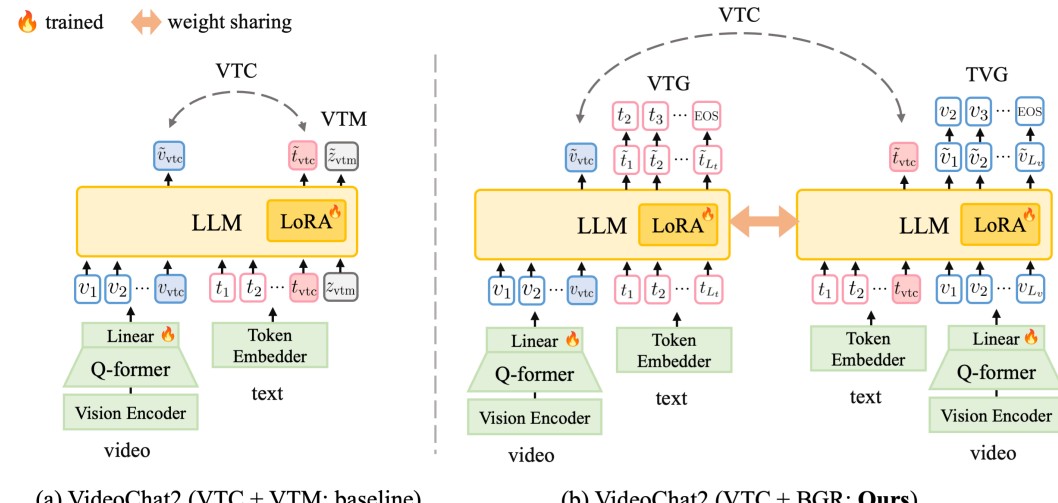

(a) VideoChat2 (VTC + VTM; baseline)     (b) VideoChat2 (VTC + BGR; **Ours**)

Figure 3: **Overall architecture.** (a) VideoChat2 with VTC and VTM (baseline). (b) VideoChat2 with VTC and our Bidirectional Text-Video Generative Retrieval (BGR), which optimizes both video-grounded text generation (VTG) and text-grounded video feature generation (TVG).

## 3.2 BIDIRECTIONAL TEXT-VIDEO GENERATIVE RETRIEVAL

Unlike the non-generative baseline of MLLM that focuses on the feature alignment using VTM, to leverage the generative approach of MLLMs, we propose to generate the other modality (*e.g.*, text) given the paired modality (*e.g.*, video) as a condition (and vice versa), namely Bidirectional Text-Video Generative Retrieval (BGR). The overall architecture of BGR is depicted in Fig. 3b.

**Training procedure.** Along with the common pretraining objective for MLLMs of video-grounded text generation (VTG), we newly introduce a text-grounded video feature generation (TVG) objective for bidirectionality. We train a model for both objectives with shared parameters. To distinguish the two different generations, we denote the probabilities measured by VTG and TVG as $P_{\text{vtg}}$ and $P_{\text{tvg}}$. Given a paired text $\mathbf{t}$ and video $\mathbf{v}$, the **VTG** objective function of BGR is formulated as:

$$\mathcal{L}_{\text{vtg}} = -\log P_{\text{vtg}}(\mathbf{t}|\mathbf{v}) = -\sum_{i=1}^{L_t} \log P_{\text{vtg}}(t_i|t_{<i}, \mathbf{v}), \tag{3}$$

where $P_{\text{vtg}}(t_i|t_{<i}, \mathbf{v}) = \text{Softmax}(\text{Linear}(\tilde{t}_{i-1}))$. $\tilde{t} \in \mathbb{R}^D$ is the output feature of $t \in \mathbb{R}^D$ and $t_0$ is the start token of the text sequence.

On the other hand, we formulate the **TVG** loss as a feature generation inspired by Fu et al. (2023b); Luo et al. (2020). The objective function of TVG is formulated as:

$$\mathcal{L}_{\text{tvg}} = -\log P_{\text{tvg}}(\mathbf{v}|\mathbf{t}) = -\sum_{i=1}^{L_v} \log \frac{\exp\left(\tilde{v}_{i-1}^{\top} v_i\right)}{\sum_{n=1}^{\mathcal{V}} \exp\left(\tilde{v}_{i-1}^{\top} v_i^{(n)}\right)}, \tag{4}$$

where $\mathcal{V}$ is the number of training videos and $\tilde{v} \in \mathbb{R}^D$ is the output feature of $v \in \mathbb{R}^D$. However, we empirically observe that generating mean-pooled features at once rather than the sequence of video tokens yields better performance, so the final TVG loss is defined as:

$$\mathcal{L}_{\text{tvg}} = -\log P_{\text{tvg}}(\mathbf{v}|\mathbf{t}) = -\log \frac{\exp\left(\tilde{v}_0^{\top} \bar{\mathbf{v}}\right)}{\sum_{n=1}^{\mathcal{V}} \exp\left(\tilde{v}_0^{\top} \bar{\mathbf{v}}^{(n)}\right)}, \tag{5}$$

where $v_0$ is the start of the video token sequence and $\bar{\mathbf{v}} \in \mathbb{R}^D$ denotes the mean-pooled feature of $\mathbf{v} \in \mathbb{R}^{L_v \times D}$. Eq. (5) encourages the model to generate the video content feature from the entire training videos $\{\mathbf{v}^{(1)}, \ldots, \mathbf{v}^{(\mathcal{V})}\}$.

Overall, we train BGR by combining the two objectives along with VTC as:

$$\mathcal{L}_{\text{bgr}} = \mathcal{L}_{\text{vtg}} + \mathcal{L}_{\text{tvg}} + \mathcal{L}_{\text{vtc}}, \tag{6}$$

where $\mathcal{L}_{\text{vtc}}$ can be obtained by Eq. (1). We note that two special tokens for VTC, $v_{\text{vtc}}$ and $t_{\text{vtc}}$, are inserted into the VTG model and TVG model respectively to reduce the number of forward processes for training efficiency.

**Inference procedure.** During inference, given the task, it is more intuitive to use the output score of the model that generates the *content* given the query as the condition, which is referred to as the *content-generation* model of BGR (*e.g.*, V2T with VTG). Yet, instead, we observe that the *query-generation* model of BGR that generates the *query* given the content as the condition (*e.g.*, V2T with TVG) outperforms the content-generation model by alleviating dependencies on priors.

Returning to the objectives of TVG and VTG in Eq. (3) and (4), each objective can be rewritten as:

$$\underbrace{P_{\text{tvg}}(\mathbf{v}|\mathbf{t})}_{\text{TVG-T2V}} \propto \underbrace{P_{\text{tvg}}(\mathbf{t}|\mathbf{v})}_{\text{TVG-V2T}} P_{\text{tvg}}(\mathbf{v}), \text{ and } \underbrace{P_{\text{vtg}}(\mathbf{t}|\mathbf{v})}_{\text{VTG-V2T}} \propto \underbrace{P_{\text{vtg}}(\mathbf{v}|\mathbf{t})}_{\text{VTG-T2V}} P_{\text{vtg}}(\mathbf{t}). \tag{7}$$

In Eq. (7), the content-generation model, *i.e.*, TVG-T2V and VTG-V2T, suffers from the prior bias, leading the model to ignore the visual-text correspondence and only take into account the content prior probability itself. In other words, the prediction of T2V using TVG is biased towards the video prior $P_{\text{tvg}}(\mathbf{v})$ in Eq. (7) (left), and the prediction of V2T with VTG is biased towards the text prior $P_{\text{vtg}}(\mathbf{t})$ in Eq. (7) (right). On the other hand, the query-generation models, *i.e.*, TVG-V2T and VTG-T2V, are less dependent on the content prior, resulting in the alleviation of prior bias. Overall, the inference procedure using both directional models for T2V and V2T is written as:

$$\mathbf{v}^* = \arg\max_{\mathbf{v}} \underbrace{P_{\text{vtg}}(\mathbf{v}|\mathbf{t}_{\text{query}})}_{\text{query-generation}} + \underbrace{P_{\text{tvg}}(\mathbf{v}|\mathbf{t}_{\text{query}})}_{\text{content-generation}}, \quad \mathbf{t}^* = \arg\max_{\mathbf{t}} \underbrace{P_{\text{tvg}}(\mathbf{t}|\mathbf{v}_{\text{query}})}_{\text{query-generation}} + \underbrace{P_{\text{vtg}}(\mathbf{t}|\mathbf{v}_{\text{query}})}_{\text{content-generation}}. \tag{8}$$

In Eq. (8), both the content-generation and query-generation models are used to find optimal content among the content candidates. In the **content-generation model**, the optimal content with the maximum likelihood is chosen by substituting the content candidates one by one in the **output** of the model, which is **most likely to be generated by the query**. On the other hand, in **the query-generation model**, the optimal content with the maximum likelihood is chosen by substituting the content candidates one by one in the **input** of the model, which is **most likely to generate the query**. For example, in T2V, for the query-generation model, we fix the output of $P_{\text{vtg}}$ as the given text query and seek the best video content which most likely generates the text query by maximizing $P_{\text{vtg}}(\mathbf{v}|\mathbf{t}_{\text{query}})$. On the other hand, for the content-generation model, we fix the input of $P_{\text{tvg}}$ as the given text query and seek the best video content which is most likely to be generated by maximizing $P_{\text{tvg}}(\mathbf{v}|\mathbf{t}_{\text{query}})$. The optimal text content is similarly retrieved in V2T. Further details of the inference strategy are provided in Sec. C. As a result, the bidirectional generation capability of BGR enables retrieval leveraging the query-generation model that easily decouples the prior probability to alleviate the MLLMs' over-reliance on the priors.

### 3.3 PRIOR NORMALIZATION

To further alleviate the prior bias at retrieval inference, we here introduce a training-free score calibration method, Prior Normalization. Also, we extend the Prior Normalization to a decoding scheme, Prior Normalized Decoding, which is applicable to a wide range of multi-modal tasks, *e.g.*, visual question answering and visual captioning.

**Prior Normalization in Text-Video Retrieval.** We aim to calibrate the probability $P(\mathbf{t}|\mathbf{v})$ by normalizing the effect of the text prior $P(\mathbf{t})$ as:

$$P(\mathbf{t}|\mathbf{v}) = \frac{P(\mathbf{v}|\mathbf{t})P(\mathbf{t})}{P(\mathbf{v})} \quad \rightarrow \quad \frac{P(\mathbf{t}|\mathbf{v})}{P(\mathbf{t})^{\alpha}} = \frac{P(\mathbf{v}|\mathbf{t})P(\mathbf{t})^{1-\alpha}}{P(\mathbf{v})}, \tag{9}$$

where $\alpha \in [0, 1]$ is a hyperparameter which determines a normalization strength. Instead of directly using the standard probability of $P(\mathbf{t}|\mathbf{v})$ in Eq. (9) (left), we normalize it with the text prior $P(\mathbf{t})$ in Eq. (9) (right). Then, the normalized probability $P^{\alpha}(\mathbf{t}|\mathbf{v})$ is defined as:

$$\log P^{\alpha}(\mathbf{t}|\mathbf{v}) \triangleq \log \frac{P(\mathbf{t}|\mathbf{v})}{P(\mathbf{t})^{\alpha}} = \log P(\mathbf{t}|\mathbf{v}) - \alpha \cdot \log P(\mathbf{t}). \tag{10}$$

$P^\alpha(\mathbf{v}|\mathbf{t})$ is similarly defined to normalize the effects of the video prior. We obtain the prior probability as $P(\mathbf{t}) \triangleq \sum_{\mathbf{v}} P(\mathbf{v}, \mathbf{t})$ by calculating all the combinations of video-text correspondences $P(\mathbf{v}, \mathbf{t})$ using $P_{\text{vtg}}$ or $P_{\text{tvg}}$. As a result, we use the normalized probability $P^\alpha(\mathbf{t}|\mathbf{v})$ to search an optimal text in Eq. (8) at inference, leading to an alleviation of biased prediction toward the prior. Interestingly, we observe that Prior Normalization is particularly effective in the content-generation model since the prior bias is prevalent in the content-generation model but already alleviated in the query-generation model (see Sec. 4.2 for details).

Prior Normalization has been actively explored in the literature, including in the context of Bayesian approach (Cui et al., 2022; Ye et al., 2022) and maximum a posterior (MAP) estimation (Doucet et al., 2002; Fei-Fei & Perona, 2005). For example, Cui et al. (2022) have proposed an elegant prior normalization technique that transforms non-Gaussian prior into standard Gaussian distributions, enabling LIS-based MCMC sampling for high-dimensional Bayesian inverse problems. However, in this work, we first study prior normalization in the context of text-video retrieval with MLLMs by interpreting the likelihood of a single modality as the prior for calculating the joint probability.

**Prior Normalized Decoding.** We now extend Prior Normalization to a decoding scheme and introduce Prior Normalized Decoding for various multi-modal tasks, *e.g.*, visual question answering and visual captioning. Instead of the standard decoding based on the probability $P(\mathbf{t}|\mathbf{v}) = \prod_i P(t_i|t_{<i}, \mathbf{v})$, we use the normalized probability $P^\alpha(\mathbf{t}|\mathbf{v})$ to decode a text sequence. Here, unlike in retrieval, we simply obtain the prior probability $P(\mathbf{t})$ in Eq. (10) with an empty video, *i.e.*, $P(\mathbf{t}) \triangleq P(\mathbf{t}|\mathbf{v}_{\text{empty}})$, for decoding efficiency. By applying our Prior Normalized Decoding to various sampling strategies (*e.g.*, nucleus sampling), the model generates the debiased text sequence by alleviating the reliance on the textual contents and utilizing the visual contents more (See Sec. 4.2 for experimental results on comprehensive image and video understanding benchmarks).

## 4 EXPERIMENTS

In this section, we evaluate the benefits of our method on four popular Text-Video Retrieval benchmark datasets and seven image and video understanding benchmarks. We first show an extensive analysis of BGR and then verify the effectiveness of Prior Normalization and Prior Normalized Decoding. Finally, we compare our model with the state-of-the-art models.

**Datasets.** For retrieval, we use DiDeMo (Anne Hendricks et al., 2017), ActivityNet (Caba Heilbron et al., 2015), LSMDC (Rohrbach et al., 2017), and MSRVTT (Xu et al., 2016) which contain diverse-length video and caption pairs. For a comprehensive evaluation of multi-modal understanding, we use MME (Fu et al., 2023a), MMBench (Liu et al., 2023c), SeedBench (Li et al., 2023a), MVBench (Li et al., 2024b), VideoMME (Fu et al., 2024), MLVU (Zhou et al., 2024), and NExT-QA (Xiao et al., 2021). Further dataset and implementation details are in Sec. A and Sec. B.

### 4.1 ANALYSIS ON BGR

**Quantitative analysis.** We first verify the effectiveness of our BGR in alleviating the prior bias derived from the LLMs' pretrained knowledge in MLLMs when conducting Text-Video Retrieval. The upper half of Tab. 1 (1) $\sim$ (4) shows the performance of BGR for both T2V and V2T on four datasets, where there exist two variant results for each model of BGR, *i.e.*, (1) VTG-V2T, (2) TVG-V2T and (3) VTG-T2V, (4) TVG-T2V. Notably, across all datasets, the content-generation model ((1) VTG-V2T and (4) TVG-T2V) is prone to exploit the prior bias during prediction, whereas the query-generation model ((2) TVG-V2T and (3) VTG-T2V) shows enhanced performance by alleviating such prior bias. For instance, in V2T, the query-generation model (TVG-V2T) in (2) shows 43.1, 39.6, 22.7, and 39.3 increase in R@1 compared to the content-generation model (VTG-V2T) in (1) on DiDeMo, ActivityNet, LSMDC, and MSRVTT, respectively. Similarly, for T2V, the query-generation model (VTG-T2V) shows 28.9, 29.4, 21.2, and 14.8 improvements over the content-generation model (TVG-T2V) by comparing (3) and (4).

Moreover, our backbone MLLM has a stronger text prior than the video prior since it is built on an LLM, so the performance gain by reducing the effect of prior is more dramatic in the VTG model with the text prior than in the TVG model with the video prior. In Tab. 1, the performance gap between T2V and V2T of the VTG (text generation) model ((1) $\rightarrow$ (3)) is larger than the one of the TVG (video feature generation) model ((4) $\rightarrow$ (2)) across all the datasets. Hence, it is crucial

Table 1: **Ablation study on BGR and Prior Normalization.** The upper half of the table (1) ∼ (4) shows the performance without Prior Normalization, and the lower half (5) ∼ (8) presents the one with Prior Normalization. Blue and red denote the content-generation models and the query-generation models, respectively.

| | DiDeMo | | | ActivtyNet | | | LSMDC | | | MSRVTT | | |
|---|---|---|---|---|---|---|---|---|---|---|---|---|
| | R@1 | R@5 | R@10 | R@1 | R@5 | R@10 | R@1 | R@5 | R@10 | R@1 | R@5 | R@10 |
| **Video-to-Text Retrieval (V2T)** | | | | | | | | | | | | |
| (1) $P_{vtg}(\mathbf{t}\|\mathbf{v})$ (VTG-V2T) | 12.4 | 52.0 | 78.7 | 14.1 | 46.0 | 73.5 | 11.0 | 36.2 | 60.4 | 11.1 | 44.6 | 72.3 |
| (2) $P_{tvg}(\mathbf{t}\|\mathbf{v})$ (TVG-V2T) | **55.5** | **83.8** | **90.9** | **53.7** | **82.8** | **91.2** | **33.7** | **57.9** | **69.0** | **50.4** | **77.1** | **85.7** |
| **Text-to-Video Retrieval (T2V)** | | | | | | | | | | | | |
| (3) $P_{vtg}(\mathbf{v}\|\mathbf{t})$ (VTG-T2V) | **76.4** | **91.2** | **93.8** | **71.0** | **90.3** | **94.8** | **45.8** | **65.1** | **73.0** | **56.6** | **79.4** | **87.0** |
| (4) $P_{tvg}(\mathbf{v}\|\mathbf{t})$ (TVG-T2V) | 47.5 | 79.0 | 89.3 | 41.6 | 77.7 | 90.3 | 24.6 | 54.2 | 65.6 | 41.8 | 73.2 | 83.5 |
| **Video-to-Text Retrieval (V2T)** | | | | | | | | | | | | |
| (5) $P_{vtg}(\mathbf{t}\|\mathbf{v})$ + **Prior Normalization** | 72.2 | 91.6 | 93.8 | 66.2 | 89.0 | 93.3 | 44.5 | 65.0 | 71.9 | 55.5 | 80.3 | 87.9 |
| (6) $P_{tvg}(\mathbf{t}\|\mathbf{v})$ + **Prior Normalization** | 56.1 | 83.7 | 90.9 | 53.8 | 82.8 | 91.2 | 34.1 | 58.2 | 69.0 | 50.5 | 76.8 | 85.8 |
| **Text-to-Video Retrieval (T2V)** | | | | | | | | | | | | |
| (7) $P_{vtg}(\mathbf{v}\|\mathbf{t})$ + **Prior Normalization** | 76.7 | 91.2 | 93.8 | 71.3 | 90.2 | 94.8 | 46.1 | 64.9 | 72.6 | 56.8 | 78.9 | 87.1 |
| (8) $P_{tvg}(\mathbf{v}\|\mathbf{t})$ + **Prior Normalization** | 56.5 | 83.6 | 90.7 | 54.9 | 84.1 | 93.0 | 32.5 | 58.5 | 69.2 | 49.6 | 76.5 | 86.2 |

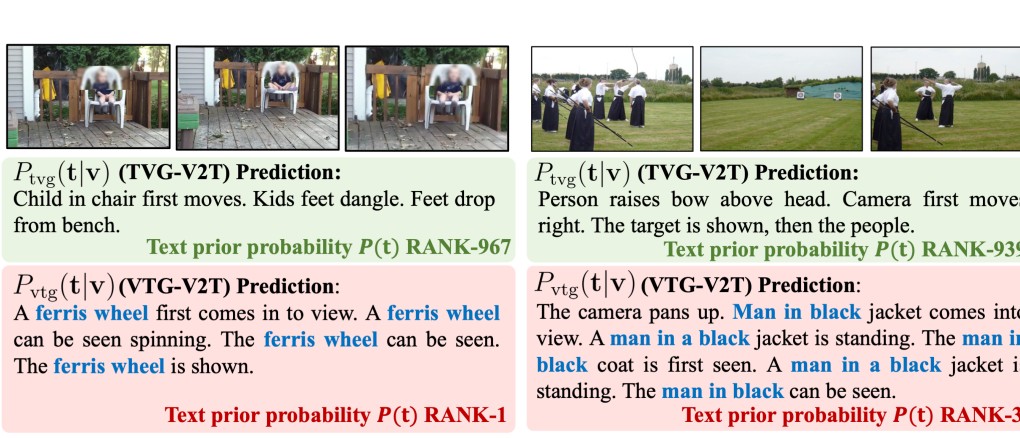

$P_{tvg}(\mathbf{t}|\mathbf{v})$ **(TVG-V2T) Prediction:**
Child in chair first moves. Kids feet dangle. Feet drop from bench.
**Text prior probability $P(\mathbf{t})$ RANK-967**

$P_{vtg}(\mathbf{t}|\mathbf{v})$ **(VTG-V2T) Prediction**:
A **ferris wheel** first comes in to view. A **ferris wheel** can be seen spinning. The **ferris wheel** can be seen. The **ferris wheel** is shown.
**Text prior probability $P(\mathbf{t})$ RANK-1**

(a)

$P_{tvg}(\mathbf{t}|\mathbf{v})$ **(TVG-V2T) Prediction:**
Person raises bow above head. Camera first moves right. The target is shown, then the people.
**Text prior probability $P(\mathbf{t})$ RANK-939**

$P_{vtg}(\mathbf{t}|\mathbf{v})$ **(VTG-V2T) Prediction**:
The camera pans up. **Man in black** jacket comes into view. A **man in a black** jacket is standing. The **man in black** coat is first seen. A **man in a black** jacket is standing. The **man in black** can be seen.
**Text prior probability $P(\mathbf{t})$ RANK-3**

(b)

Figure 4: **Examples of V2T predictions with BGR.** Green signifies the accurate prediction, while red denotes the incorrect prediction. The blue text stands for the repeated phrases.

to mitigate the model's high reliance on the prior during inference and we show that BGR is an effective method where the simple adoption of the query-generation model plays a pivotal role in ensuring that the model is not susceptible to the prior bias in its predictions.

**Qualitative analysis.** We here analyze the effectiveness of BGR on V2T with qualitative results in Fig. 4. We observe that the query-generation model (TVG-V2T) successfully retrieves the ground-truth caption from the given video, but the content-generation model (VTG-V2T) retrieves the incorrect caption which totally ignores the correspondence with the given video. This phenomenon of retrieving irrelevant captions indicates that the model prioritizes captions with high ranks in terms of the text prior probability. For example, in Fig. 4a, the text prior probability rank of the ground-truth caption is 967 out of 1,003, while the caption predicted by VTG is 1. Similarly, the text prior probability rank of the ground-truth caption in Fig. 4b is 939 out of 1,003, whereas the one of the caption predicted by the VTG model is 3. Interestingly, we also observe that the high text prior probability is assigned to the caption which is longer and contains repeated phrases (*e.g.*, "ferris wheel" in Fig. 4a and "man in black" in Fig. 4b) due to the autoregressive property of LLMs. The caption with a high text prior probability hinders the accurate retrieval in VTG-V2T, while the prior bias is successfully mitigated in TVG-V2T. Therefore, adopting the query-generation model of BGR is effective in that it deviates from prior bias which degrades the performance.

Table 2: **Results of Prior Normalized Decoding on various multi-modal tasks.** The performances on seven different benchmarks are reported. † stands for the model with Prior Normalized Decoding.

| Model | Image Understanding Benchmark | | | | Video Understanding Benchmark | | | | | | avg. $\Delta$ |
|---|---|---|---|---|---|---|---|---|---|---|---|
| | MME | | MMBench | SeedBench | MVBench | VideoMME | | MLVU | NExT-QA | SeedBench | |
| | perception | cognition | en-dev | image | test | w/o subtitle | w/ subtitle | m-avg | mc-val | video | |
| GPT-4V Achiam et al. (2023) | 1409.0 | 517.0 | 75.0 | 49.9 | 43.5 | 59.9 | 63.3 | 49.2 | - | 60.5 | - |
| VILA (Lin et al., 2024) | 1762.0 | | 82.4 | 75.8 | - | 60.1 | 61.1 | - | 67.9 | - | - |
| IXC-2.5 (Zhang et al., 2024) | 2229.0 | | 82.2 | 75.4 | 69.1 | 55.8 | 58.8 | 37.3 | 71.0 | - | - |
| VideoChat2 (Li et al., 2024b) | 1231.4 | 274.3 | 63.9 | 67.8 | 60.1 | 42.2 | 53.0 | 45.8 | 78.9 | 54.5 | - |
| VideoChat2† (Ours) | **1284.5** | **322.5** | **66.2** | **68.0** | **62.3** | **47.1** | **56.3** | **48.5** | **79.4** | **55.4** | **+11.8** |
| LLaVA-Onevision (Li et al., 2024a) | 1696.7 | 514.6 | 79.8 | 75.0 | 57.1 | 58.5 | 57.8 | 65.3 | 79.4 | 56.9 | - |
| LLaVA-Onevision† (Ours) | **1708.6** | **535.0** | **81.3** | **75.3** | **58.9** | **61.7** | **62.1** | **65.8** | **79.5** | **57.0** | **+4.4** |
| InternVL2 (Chen et al., 2024b) | 1622.7 | 582.5 | 81.8 | 76.1 | 65.8 | 51.3 | 51.7 | 50.8 | 80.4 | 56.4 | - |
| InternVL2† (Ours) | **1642.1** | **590.0** | **82.7** | **76.2** | **67.1** | **54.7** | **55.1** | **55.1** | **80.8** | **56.6** | **+4.1** |

## 4.2 ANALYSIS ON PRIOR NORMALIZATION

**Analysis of Prior Normalization on Retrieval.** We verify the effectiveness of Prior Normalization on Text-Video Retrieval shown in the lower half of Tab. 1 (5) $\sim$ (8). We observe that the performance of the content-generation model (TVG-T2V and VTG-V2T) is significantly improved after applying our Prior Normalization. Specifically, in VTG-V2T, by comparing (1) and (5), the R@1 gain is 59.8, 52.1, 33.5, and 44.4 on each dataset. The efficacy of Prior Normalization is notable especially in the content-generation models since simple adoption of the query-generation models of our BGR has already alleviated the prior bias. Therefore, in terms of the query-generation models ((2) → (6) and (3) → (7)), they show a relatively small performance gain or even on par with models without Prior Normalization. Furthermore, by comparing (1) and (5), the performance gain on VTG-V2T is relatively larger than the one on TVG-T2V ((4) and (8)) even though both are content-generation models, attributed to strong text prior bias on V2T than the video prior bias on T2V due to the high reliance of MLLMs on LLMs' knowledge. Finally, after Prior Normalization, the VTG models outperform the TVG models both in V2T and T2V since most MLLMs are pretrained only with the VTG loss and the TVG loss is newly introduced at the fine-tuning phase.

**Analysis of Prior Normalized Decoding on various multi-modal understanding benchmarks.** We provide an in-depth analysis of Prior Normalized Decoding on multi-modal understanding benchmarks beyond retrieval. Tab. 2 presents the evaluation results on seven image and video understanding benchmarks which contain comprehensive tasks evaluating the model's image/video understanding and reasoning ability. By applying Prior Normalized Decoding to three different MLLMs (VideoChat2 (Li et al., 2024b), LLaVA-Onevision (Li et al., 2024a), and InternVL2 (Chen et al., 2024b)), the performances are consistently improved across all the benchmarks by a margin of 11.8, 4.4, and 4.1 on average, respectively. This result implies that our simple training-free score calibration for decoding successfully alleviates the MLLMs' over-reliance on the text and improves the quality of the generated output by encouraging the models to refer more to the visual information.

## 4.3 COMPARISON WITH STATE-OF-THE-ART MODELS

In Tab. 3, we compare our model with state-of-the-art models both on T2V and V2T. First, by comparing UMT and VideoChat2 baseline with the VTM loss, introduced in Sec. 3.1, the LLMs-based VideoChat2 underperforms the non-LLMs-based UMT in 5 out of 8 settings. However, with our generative VTG objective, the model outperforms the VTM model across all settings. Also, since a notable performance gain is observed in the content-generation models after applying Prior Normalization as in Sec. 4.2, using an ensemble of the content-generation and the query-generation model, BGR (*i.e.*, VTG + TVG), surpasses the VTM model by a margin of 6.6 in R@1 on average. Finally, the ensemble of the model's output score optimized over three different objectives denoted as BGR* (VTG + TVG + VTM), achieves the best performance across all datasets. Specifically, in DiDeMo and ActivityNet, the gap between our final model and UMT is 12.0 and 9.1 respectively in T2V. As a result, our model achieves state-of-the-art performances both in V2T and T2V with an 8.3 increase in R@1 on average compared to UMT. More detailed results are provided in Sec. D.

Table 3: **Comparison with state-of-the-art models.** Prior Normalization is applied to our model variants colored in blue. We report R@1 both on T2V and V2T.

| | DiDeMo | | ActivityNet | | LSMDC | | MSRVTT | |
|---|---|---|---|---|---|---|---|---|
| | T2V | V2T | T2V | V2T | T2V | V2T | T2V | V2T |
| All-in-one (Wang et al., 2023) | 32.7 | - | 22.4 | - | - | - | 37.9 | - |
| LAVENDER (Li et al., 2023e) | 53.4 | - | - | - | 26.1 | - | 40.7 | - |
| VINDLU (Cheng et al., 2023) | 61.2 | 61.2 | 55.0 | - | - | - | 46.5 | - |
| CLIP4Clip (Luo et al., 2022) | 42.8 | 42.5 | 40.5 | 42.6 | 21.6 | 20.9 | 44.5 | 43.1 |
| CLIP-ViP (Xue et al., 2023) | 50.5 | - | 53.4 | - | 29.4 | - | 54.2 | - |
| Cap4Video (Wu et al., 2023) | 52.0 | - | - | - | - | - | 51.4 | 49.0 |
| T-MASS (Wang et al., 2024) | 55.3 | - | - | - | 30.3 | - | 52.7 | 50.9 |
| MV-Adapter (Jin et al., 2024) | 44.3 | 42.7 | 42.9 | 43.6 | 23.2 | 24.0 | 46.2 | 47.2 |
| TeachCLIP (Tian et al., 2024) | 43.7 | - | 42.2 | - | - | - | 46.8 | - |
| InternVideo (Wang et al., 2022b) | 57.9 | 59.1 | 62.2 | 62.8 | 34.0 | 34.9 | 55.2 | 57.9 |
| UMT (Li et al., 2023d) | 70.4 | 67.9 | 66.8 | 64.4 | 43.0 | 41.4 | 58.8 | 58.6 |
| **VideoChat2** | | | | | | | | |
| VTM (Baseline) | 74.6 | 67.1 | 63.0 | 54.1 | 44.7 | 44.4 | 53.0 | 52.9 |
| VTG (**Ours**) | 76.7 | 72.2 | 71.3 | 66.2 | 46.1 | 44.5 | 56.8 | 55.5 |
| TVG (**Ours**) | 56.5 | 56.1 | 54.9 | 53.8 | 32.5 | 34.1 | 49.6 | 50.5 |
| BGR (**Ours**) | 78.6 | 74.0 | 73.6 | 69.5 | 47.7 | 44.9 | 59.8 | 58.7 |
| BGR$^*$ (**Ours**) | **82.4** | **80.2** | **75.9** | **73.9** | **51.1** | **50.9** | **62.0** | **61.2** |

## 5 RELATED WORKS

**Multi-modal Large Language Models (MLLMs).** In recent years, there has been a significant surge in the development of large language models (LLMs) (Chowdhery et al., 2023; Brown et al., 2020; Touvron et al., 2023; Chiang et al., 2023; Achiam et al., 2023; Jiang et al., 2023; Chung et al., 2022). Following the achievements, integrating a visual encoder into such LLMs has propelled progress in the vision-language multi-modal domain, known as multi-modal large language models (MLLMs) (Alayrac et al., 2022; Driess et al., 2023; Liu et al., 2023b; Li et al., 2023b; 2024b; Zhu et al., 2023). A notable MLLM includes Flamingo (Alayrac et al., 2022), which aligns the pretrained vision encoder and the language model and handles a sequence of text tokens interleaved with the visual content. More recently, both image MLLMs (Liu et al., 2023b; Zhang et al., 2023b; Liu et al., 2023a; Zhu et al., 2023) and video MLLMs (Li et al., 2023c; 2024b; Maaz et al., 2023; Zhang et al., 2023a) are further trained by visual instruction tuning to enhance visual understanding and reasoning. For example, VideoChat2 (Li et al., 2024b) has achieved state-of-the-art performances on various video multi-modal tasks by 3-stage pretraining with diverse instruction-tuning data.

**Text-Video Retrieval.** Text-Video Retrieval is one of the popular multi-modal tasks that aims to find the most relevant video based on text or vice versa. Early studies (Luo et al., 2022; Gorti et al., 2022; Liu et al., 2022; Wu et al., 2023; Xue et al., 2023) have focused on leveraging pretrained CLIP (Radford et al., 2021) for joint embedding space learning between video and text. For example, Wu et al. (2023) have proposed Cap4Video by utilizing captions to enhance video-text alignment based on CLIP. However, such studies have shown unsatisfactory performances due to the limitation of shallow fusion of the dual encoders. Recently, large-scale multi-modal video foundation models (Cheng et al., 2023; Wang et al., 2022a; Li et al., 2023d) have boosted the performance by implementing a cross-modal fusion, which learns the fine-grained interactions between the two modalities. Those models have adopted a hybrid inference approach which efficiently retrieves the top-*k* candidates by VTC and then computes their pairwise scores by VTM for more accurate retrievals.

## 6 CONCLUSION

In this paper, we observe that content-generation with MLLMs tends to retrieve the incorrect caption from a given video (and vice versa) due to the prior bias in Text-Video Retrieval. Therefore, we propose a Bidirectional Text-Video Generative Retrieval (BGR) to address the over-reliance on the prior by ensembling predictions from both directions, *i.e.*, content-generation and query-generation. Moreover, our simple plug-and-play score calibration module, Prior Normalization, further improves the performances along with BGR in Text-Video Retrieval by reducing dependence on the prior. Our experimental results highlight the potential of Prior Normalized Decoding applied to MLLMs in a more balanced consideration of both textual and visual information in various multi-modal tasks.

## REPRODUCIBILITY

We describe details of BGR in Sec. 3.2 and Prior Normalization in Sec. 3.3. We further provide dataset details and implementation details in Sec. A and Sec. B, respectively. Also, the inference strategy of our BGR is explained in Sec. C for reproducibility.

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

# APPENDIX

The appendix is organized into the following sections:

## A  DATASET DETAILS

### A.1  TEXT-VIDEO RETRIEVAL

**DiDeMo (Anne Hendricks et al., 2017).** Distinct Describable Moments (DiDeMo) contains 10K videos which are divided into 5-second segments. It has a total of 26K moments whose descriptions are detailed and contain camera movement, temporal transition indicators, and activities. We follow the previous works Bain et al. (2021); Lei et al. (2021); Luo et al. (2022); Wu et al. (2023); Cheng et al. (2023); Li et al. (2023d) by concatenating all captions of one video and solving the task as a paragraph-video retrieval task. The number of training and test samples is 8,394 and 1,003, respectively.

**ActivityNet (Caba Heilbron et al., 2015).** ActivityNet dataset contains 19K videos from YouTube, which are categorized into 200 different types of activities. On average, each category has 137 videos and each video has 1.41 activities which are annotated with temporal boundaries. Similar to DiDeMo, we also concatenate all the captions of a video to form a paragraph-video retrieval task on the 'val1' split by following Zhang et al. (2018); Gabeur et al. (2020); Luo et al. (2022); Cheng et al. (2023); Li et al. (2023d). Therefore, the number of training and test samples is 10,009 and 4,917, respectively.

**LSMDC (Rohrbach et al., 2017).** Large Scale Movie Description Challenge (LSMDC) contains 118K short video clips from 202 movies with captions from the movie script or from transcribed DVS (descriptive video services) for the visually impaired. Our model is trained with 101,055 videos and evaluated on 1,000 videos.

**MSRVTT (Xu et al., 2016).** Microsoft Research Video to Text (MSRVTT) contains 10K video clips from 20 categories, with each video clip annotated with 20 sentences. There are 29K unique words in all captions. Following the literature Yu et al. (2018); Luo et al. (2022); Miech et al. (2019); Gabeur et al. (2020); Wu et al. (2023); Cheng et al. (2023); Li et al. (2023d), we train our model with $9,000 \times 20$ training samples and 1,000 test samples.

### A.2  COMPREHENSIVE IMAGE AND VIDEO UNDERSTANDING

**MME (Fu et al., 2023a).** Multi-modal large language Model Evaluation benchmark (MME) is composed of 14 subtasks where all the samples are manually annotated. MME targets to assess MLLMs' perception and cognition abilities including OCR, existence of objects, commonsense reasoning, numerical calculation, code reasoning, etc.

**MMBench (Liu et al., 2023c).** MMBench is a bilingual benchmark to evaluate the MLLMs' multi-modal understanding abilities. This benchmark includes multiple-choice questions across the 20 ability dimensions like spatial relationship, physical property, attribute recognition, object localization, etc.

**SeedBench (Li et al., 2023a).** SeedBench aims at a comprehensive assessment of generative models and contains 19K manually annotated multiple-choice questions across the 12 ability dimensions both on the image and video domain. The questions cover both spatial and temporal understanding like scene understanding, action prediction, procedure understanding, etc.

**MVBench (Li et al., 2024b).** Multi-modal Video understanding Benchmark (MVBench) consists of 20 challenging video understanding tasks that can effectively assess the ability to comprehend temporal evolution in dynamic videos. It consists of 9 main tasks for spatial understanding, which are then further split into a total of 20 tasks for temporal understanding.

**VideoMME (Fu et al., 2024).** Multi-Modal Evaluation benchmark of MLLMs in Video analysis (VideoMME) evaluates the ability of MLLMs to handle sequential visual data on 6 primary visual domains with 30 subcategories. The videos are categorized as short, medium, and long, ranging from 11 seconds to 1 hour. A total of 900 videos are in the benchmark with 2,700 questions.

**MLVU (Zhou et al., 2024).** Multi-task Long Video Understanding benchmark (MLVU) targets to assess long video understanding performance spanning 7 video genres including movies, egocentric videos, cartoons, etc. MLVU contains 2,593 questions on 9 categories like topic reasoning, plot question answering, action count, ego reasoning, etc.

**NExT-QA (Xiao et al., 2021).** NExT-QA is a video question answering task aiming to evaluate causal action reasoning, temporal action reasoning, and common scene comprehension. This dataset includes 47,692 multiple-choice questions and 52,044 open-ended questions on a total of 5,440 videos.

## B  IMPLEMENTATION DETAILS

**BGR details.** Our BGR is built upon VideoChat2 (Li et al., 2024b) and is further fine-tuned on each Text-Video Retrieval dataset. Specifically, VideoChat2 consists of a visual encoder, Q-Former, a linear projection layer, and a LLM. The visual encoder and LLM are initialized with UMT-L (Li et al., 2023d) and Mistral-7B-Instruct-v0.2 (Jiang et al., 2023), respectively. We freeze parameters in the visual encoder, Q-Former, and LLM, and only update parameters in the linear projection layer and LoRA for parameter-efficient fine-tuning, resulting in 48M trainable parameters among 7B total parameters (0.7%). We accumulate gradients from the TVG and VTG loss and update the trainable parameters at once.

Table 4: **Experimental settings in Text-Video Retrieval.**

|  | DiDeMo | ActivityNet | LSMDC | MSRVTT |
|---|---|---|---|---|
| optimizer | | AdamW | | |
| optimizer momentum | | $\beta_1 = 0.9, \beta_2 = 0.95$ | | |
| weight decay | | 1.0 | | |
| warmup epochs | | 1 | | |
| total epochs | | 5 | | |
| input frames | | 16 | | |
| max sequence length | 196 | 196 | 128 | 128 |
| learning rate | 2e-4 | 1e-4 | 1e-4 | 1e-5 |
| batch size | 20 | 18 | 32 | 36 |

**Experimental settings.** The self-attention mechanism in our model is implemented under FlashAttention2 (Dao, 2024) and we sample 16 frames per video for all datasets. As for a max sequence length, we use 128 for LSMDC and MSRVTT, and 196 for DiDeMo and ActivityNet. The learning rate is 1e-4 for ActivityNet and LSMDC, 1e-4 for MSRVTT, and 2e-4 for DiDeMo with AdamW optimizer. We train our model 5 epochs on $8 \times$ A100 GPUs with a batch size of 20, 18, 32, and

36 for DiDeMo, ActivityNet, LSMDC, and MSRVTT, respectively. For inference, we select top-16 candidates according to the similarity from the VTC loss and re-rank them by comparing their VTG (or TVG) scores. More details are summarized in Tab. 4.

## C  INFERENCE DETAILS OF BGR

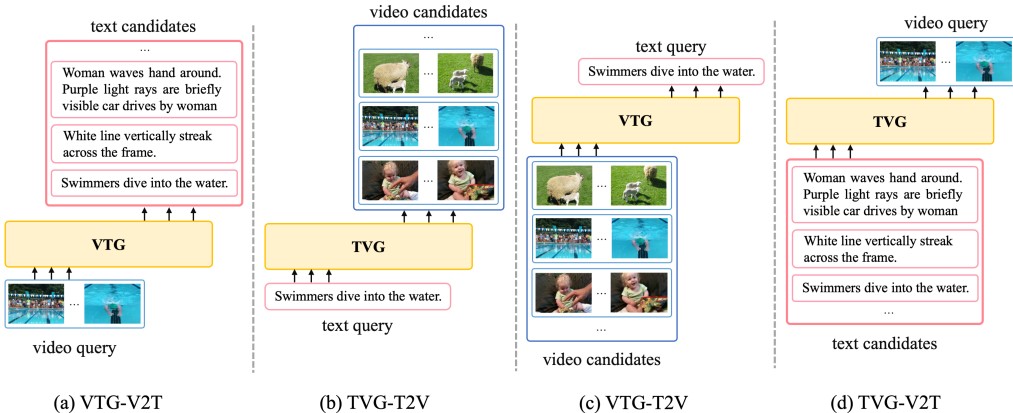

|                |                  |                |                |
|:--------------:|:----------------:|:--------------:|:--------------:|
| (a) VTG-V2T | (b) TVG-T2V | (c) VTG-T2V | (d) TVG-V2T |

Figure 5: **Four variants of BGR for inference.** Note (a) and (b) are the content-generation models of BGR, and (c) and (d) are the query-generation models of BGR.

At inference of BGR, instead of autoregressively generating the output for retrieval, BGR calculates the posterior probability in Eq. (8) by replacing the input or output of the probability with content candidates. Fig. 5 illustrates the inference details of BGR. For the content-generation models, in Fig. 5a (V2T) and Fig. 5b (T2V), we fix the *input* of the model as a video (or text) query and seek the best text (or video) content by replacing the *output* with text (or video) candidates. On the other hand, for the query-generation models, in Fig. 5c (T2V) and Fig. 5d (V2T), we fix the *output* of the model as a text (or video) query and seek the best video (or text) content by replacing the *input* with video (or text) candidates. This simple trick reduces the computational cost for generation by 72 times per text-video pair. As mentioned in Sec. 3.2, adopting the query-generation model at inference is effective in alleviating the prior bias.

## D  DETAILED RESULTS OF COMPARISON WITH STATE-OF-THE-ART MODELS

Table 5: **Comparison with state-of-the-art models on T2V.** Prior Normalization is applied to our model variants highlighted in blue cells.

|  | DiDeMo | | | ActivityNet | | | LSMDC | | | MSRVTT | | |
|---|---|---|---|---|---|---|---|---|---|---|---|---|
|  | R@1 | R@5 | R@10 | R@1 | R@5 | R@10 | R@1 | R@5 | R@10 | R@1 | R@5 | R@10 |
| **Text-to-Video Retrieval (T2V)** | | | | | | | | | | | | |
| All-in-one (Wang et al., 2023) | 32.7 | 61.4 | 73.5 | 22.4 | 53.7 | 67.7 | - | - | - | 37.9 | 68.1 | 77.1 |
| LAVENDER (Li et al., 2023e) | 53.4 | 78.6 | 85.3 | - | - | - | 26.1 | 46.4 | 57.3 | 40.7 | 66.9 | 77.6 |
| VINDLU (Cheng et al., 2023) | 61.2 | 85.8 | 91.0 | 55.0 | 81.4 | 89.7 | - | - | - | 46.5 | 71.5 | 80.4 |
| CLIP4Clip (Luo et al., 2022) | 42.8 | 68.5 | 79.2 | 40.5 | 72.4 | 83.4 | 21.6 | 41.8 | 49.8 | 44.5 | 71.4 | 81.6 |
| CLIP-ViP (Xue et al., 2023) | 50.5 | 78.4 | 87.1 | 53.4 | 81.4 | 90.0 | 29.4 | 50.6 | 59.0 | 54.2 | 77.2 | 84.8 |
| Cap4Video (Wu et al., 2023) | 52.0 | 79.4 | 87.5 | - | - | - | - | - | - | 51.4 | 75.7 | 83.9 |
| T-MASS (Wang et al., 2024) | 55.3 | 80.1 | 87.7 | - | - | - | 30.3 | 52.2 | 61.3 | 52.7 | 77.1 | 85.6 |
| MV-Adapter (Jin et al., 2024) | 44.3 | 72.1 | 80.5 | 42.9 | 74.5 | 85.7 | 23.2 | 43.9 | 53.2 | 46.2 | 73.2 | 82.7 |
| TeachCLIP (Tian et al., 2024) | 43.7 | 71.2 | - | 42.2 | 72.7 | - | - | - | - | 46.8 | 74.3 | - |
| InternVideo (Wang et al., 2022b) | 57.9 | 82.4 | 88.9 | 62.2 | 85.9 | 93.2 | 34.0 | 53.7 | 62.9 | 55.2 | 79.6 | 87.5 |
| UMT (Li et al., 2023d) | 70.4 | 90.1 | 93.5 | 66.8 | 89.1 | 94.9 | 43.0 | 65.5 | 73.0 | 58.8 | 81.0 | 87.1 |
| UMT + **Prior Normalization (Ours)** | **75.4** | **92.6** | **94.9** | **73.2** | **91.5** | **96.0** | **45.1** | **67.2** | **74.6** | **62.4** | **83.5** | **88.1** |
| **VideoChat2** | | | | | | | | | | | | |
| VTM (Baseline) | 74.6 | 90.3 | 93.4 | 63.0 | 87.7 | 94.1 | 44.7 | 67.5 | 73.4 | 53.0 | 77.7 | 87.1 |
| VTG **(Ours)** | 76.7 | 91.2 | 93.8 | 71.3 | 90.2 | 94.8 | 46.1 | 64.9 | 72.6 | 56.8 | 78.9 | 87.1 |
| TVG **(Ours)** | 56.5 | 83.6 | 90.7 | 54.9 | 84.1 | 93.0 | 32.5 | 58.5 | 69.2 | 49.6 | 76.5 | 86.2 |
| BGR **(Ours)** | 78.6 | 92.6 | **93.9** | 73.6 | 91.5 | **95.1** | 47.7 | 66.6 | 73.1 | 59.8 | 80.4 | 88.0 |
| BGR* **(Ours)** | **82.4** | **93.0** | 93.8 | **75.9** | **91.9** | **95.1** | **51.1** | **68.7** | **74.1** | **62.0** | **82.4** | **88.2** |

Table 6: **Comparison with state-of-the-art models in V2T.** Prior Normalization (PN) is applied to our model variants highlighted in blue cells.

| | DiDeMo | | | ActivityNet | | | LSMDC | | | MSRVTT | | |
|---|---|---|---|---|---|---|---|---|---|---|---|---|
| | R@1 | R@5 | R@10 | R@1 | R@5 | R@10 | R@1 | R@5 | R@10 | R@1 | R@5 | R@10 |
| **Video-to-Text Retrieval (V2T)** | | | | | | | | | | | | |
| CLIP4Clip (Luo et al., 2022) | 42.5 | 70.6 | 80.2 | 42.6 | 73.4 | 85.6 | 20.9 | 40.7 | 49.1 | 43.1 | 70.5 | 81.2 |
| Cap4Video (Wu et al., 2023) | - | - | - | - | - | - | - | - | - | 49.0 | 75.2 | 85.0 |
| T-MASS (Wang et al., 2024) | - | - | - | - | - | - | - | - | - | 50.9 | 80.2 | 88.0 |
| MV-Adapter (Jin et al., 2024) | 42.7 | 73.0 | 81.9 | 43.6 | 75.0 | 86.5 | 24.0 | 42.8 | 52.1 | 47.2 | 74.8 | 83.9 |
| InternVideo Wang et al. (2022b) | 59.1 | 81.8 | 89.0 | 62.8 | 86.2 | 93.3 | 34.9 | 54.6 | 63.1 | 57.9 | 79.2 | 86.4 |
| UMT (Li et al., 2023d) | 67.9 | 88.6 | 93.0 | 64.4 | 89.1 | 94.8 | 41.4 | 64.3 | 71.5 | 58.6 | 81.6 | 86.5 |
| UMT + **Prior Normalization (Ours)** | **75.7** | **92.3** | **95.5** | **73.8** | **92.1** | **96.5** | **45.7** | **67.9** | **73.9** | **63.0** | **83.8** | **89.1** |
| **VideoChat2** | | | | | | | | | | | | |
| VTM (Baseline) | 67.1 | 89.9 | 93.5 | 54.1 | 84.1 | 91.7 | 44.4 | 65.1 | 72.0 | 52.9 | 78.5 | 87.2 |
| VTG **(Ours)** | 72.2 | 91.6 | 93.8 | 66.2 | 89.0 | 93.3 | 44.5 | 65.0 | 71.9 | 55.5 | 80.3 | 87.9 |
| TVG **(Ours)** | 56.1 | 83.7 | 90.9 | 53.8 | 82.8 | 91.2 | 34.1 | 58.2 | 69.0 | 50.5 | 76.8 | 85.8 |
| BGR **(Ours)** | 74.0 | 92.1 | **93.8** | 69.5 | 90.2 | 93.3 | 44.9 | 65.8 | 72.6 | 58.7 | 81.7 | 88.2 |
| BGR* **(Ours)** | **80.2** | **92.8** | 93.7 | **73.9** | **90.5** | **93.6** | **50.9** | **68.8** | **73.5** | **61.2** | **82.7** | **88.4** |

Tab. 3 only demonstrates R@1 performance. We here show detailed results including R@1, R@5, and R@10 on DiDeMo, ActivityNet, LSMDC, and MSRVTT in Tab. 5 and Tab. 6. We first observe that the previous non-MLLMs-based Text-Video Retrieval model like UMT also suffers from the prior bias that hinders correct predictions. Therefore, a remarkable performance gain is observed by applying our Prior Normalization to UMT. Specifically, on T2V in Tab. 5, the R@1 is increased by 5.0, 6.4, 2.1, and 3.6 on DiDeMo, ActivityNet, LSMDC, and MSRVTT, respectively, resulting in 4.3 increase on average. However, by referring to Tab. 1, the R@1 is increased by 9.5 on average in the content-generation model. Similarly, on V2T in Tab. 6, the average R@1 gap is 6.5 in UMT while the one is 47.5 in Tab. 1. This signifies that the prior bias problem is more severe in the MLLMs-based model than in the non-MLLMs-based model, and our BGR and Prior Normalization successfully alleviate such bias especially attributed to LLMs' pretrained knowledge.

# E    FURTHER DISCUSSION ON PRIOR NORMALIZATION

## E.1    ALLEVIATION OF PRIOR BIAS

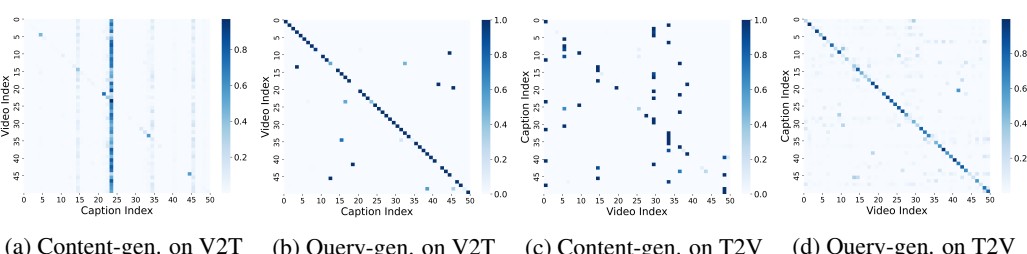

(a) Content-gen. on V2T    (b) Query-gen. on V2T    (c) Content-gen. on T2V    (d) Query-gen. on T2V

Figure 6: **Visualization of retrieval results on the content-generation and query-generation models in DiDeMo.** 50 text-video pairs are sampled to avoid visual clutter.

To verify the alleviation of the prior bias, we provide heatmaps in Fig. 6 when applying the content-generation (blue) and query-generation (orange) models. In Fig. 6a, with the content-generation model, the caption with the highest text prior probability, *i.e.*, the 24-th caption, is retrieved by most videos. On the other hand, the query-generation model leads to a balanced prediction where each caption is retrieved by its own paired video in Fig. 6b. This reveals that the query-generation model successfully alleviates the prior bias and considers video-text correspondences more. Furthermore, as discussed in Sec. 1 and Sec. 2.2, even though there exist both text prior bias and video prior bias, the text prior bias is more severe due to the high reliance of MLLMs on LLMs' pretrained knowledge. This becomes evident when comparing Fig. 6a and Fig. 6c, a clear vertical line is observed on V2T in Fig. 6a. This result indicates that MLLMs are more biased toward text prior than video prior due to the LLMs' pretrained knowledge.

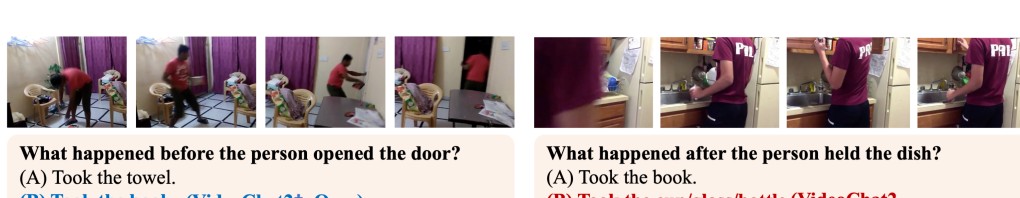

(a) **Bias towards repeated phrases.**  (b) **Bias towards common sense.**

Figure 7: **Qualitative results of Prior Normalized Decoding on MVBench.** Blue signifies the accurate prediction, while red denotes the incorrect prediction.

### E.2    Qualitative results of Prior Normalized Decoding on MVBench.

To show how Prior Normalized Decoding corrects the model's output, we demonstrate the qualitative results in Fig. 7, which illustrates predictions of VideoChat2, VideoChat2 w/o video, and VideoChat2 + Prior Normalized Decoding (*i.e.*, VideoChat2$^\dagger$). We observe that the naive VideoChat2 often tends to follow the predictions based on the text prior (*i.e.*, VideoChat2 w/o video) and outputs incorrect answers. We here discern two kinds of bias: bias towards *repeated phrases*, already discussed in Sec. 4.1, and bias towards *common sense*. In Fig. 7a, for the question of "What happened before the person opened the door?", due to the repetition of 'Opened the door' from the question in one of the answer candidates, a high text prior probability is assigned to the option "(C) Opened the door". Similarly, in Fig. 7b, for the question of "What happened after the person held the dish?", the VideoChat2 w/o video model prioritizes the likely action sequence "(B) Took the cup/glass/bottle" based on the common sense from LLMs' pretrained knowledge where one usually takes the cup/glass/bottle after holding the dish. Hence, VideoChat2 also predicts an incorrect answer by highly depending on the wrong text prior while our Prior Normalized Decoding successfully corrects the answer by reducing the effect of text prior and referring more to the visual information. Overall, Prior Normalized Decoding is a simple task-agnostic score calibration method that alleviates the predominance of linguistic cues and ensures balanced consideration between the video and text.

### E.3    Prior Normalized Decoding in Visual Captioning

We also apply our Prior Normalized Decoding to VideoChat2 on video captioning in Fig. 8. The standard VideoChat2 usually generates a hallucinated text by overlooking the visual content. For example, in Fig. 8a, the word 'apple' is hallucinated which does not appear in the video. Similarly, in Fig. 8b, the standard VideoChat2 also generates a hallucinated phrase "They are trimming the dog's nails" while the dog licks his feet in the video. However, with our Prior Normalized Decoding (denoted as VideoChat2$^\dagger$), the hallucinated text is successfully removed by encouraging the model to take into account visual contents more.

## F    Mathematical analysis on Prior Normalization

In this section, we provide a mathematical analysis on Prior Normalization. We focus on searching the most relevant video given a text query, *i.e.*, text-to-video retrieval by $P(\mathbf{v}|\mathbf{t})$, which can be written as:

$$P(\mathbf{v}|\mathbf{t}) = \frac{P(\mathbf{t}|\mathbf{v})P(\mathbf{v})}{P(\mathbf{t})}. \tag{11}$$

As discussed in Tab. 1 and Fig. 6, query generation (or likelihood) $P(\mathbf{t}|\mathbf{v})$ is more accurate than content generation $P(\mathbf{v}|\mathbf{t})$. In other words, even if $P(\mathbf{t}|\mathbf{v})$ predicts accurately, the final result $P(\mathbf{v}|\mathbf{t})$ can lead to incorrect retrievals due to a highly skewed video prior probability $P(\mathbf{v})$, namely, prior bias.

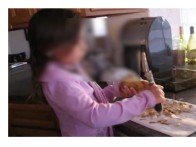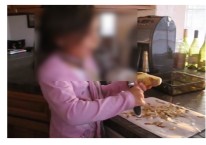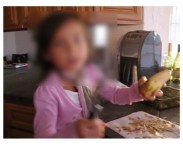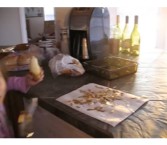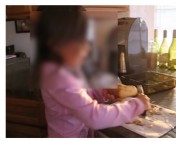

**VideoChat2:** A little girl peels an apple with an apple peeler. She cuts the apple into slices. She holds a slice up to show the camera.

**VideoChat2†:** A young girl peels potatoes on a cutting board behind a counter. The girl moves the potato across the board to get at the skin to peel it off. The girl then repeats the process to get the potato completely clean.

(a)

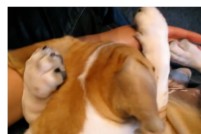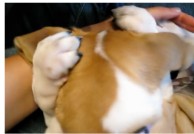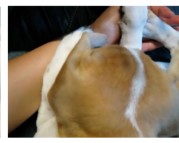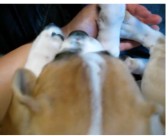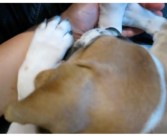

**VideoChat2:** A person is holding a little dog. They are trimming the dog's nails. The dog gets up and pants a lot.

**VideoChat2†:** A person is holding a little dog in their hands. The dog licks his feet while the person continues to hold him.

(b)

Figure 8: **Qualitative results of Prior Normalized Decoding in Video Captioning on ActivityNet.** † stands for the model with Prior Normalized Decoding. The hallucinated text is highlighted in red.

We will discuss the case where a too large prior gap in $P(\mathbf{v})$ leads to incorrect retrieval results in the posterior $P(\mathbf{v}|\mathbf{t})$ when the likelihood $P(\mathbf{t}|\mathbf{v})$ is correct but the gap is relatively small. Specifically, we first assume that the likelihood predicts correctly but the prior gap is too larger than the likelihood gap. Then we prove that the posterior probability results in a wrong prediction.

Assume that the likelihood predicts correctly, *i.e.*, $P(\mathbf{t}_m|\mathbf{v}_m) > P(\mathbf{t}_m|\mathbf{v}_n) > 0, \forall n \neq m$ but the gap is small. Then, it can be written as:

$$0 < \log P(\mathbf{t}_m|\mathbf{v}_m) - \log P(\mathbf{t}_m|\mathbf{v}_n) < \varepsilon, \tag{12}$$

where the $\varepsilon > 0$ is the maximum gap. Eq. (12) means the retrieval score of $m$-th video (positive sample) is larger than the one of $n$-th video (negative sample) given the $m$-th text, and the gap is smaller than the maximum gap $\varepsilon$.

Another assumption is that the strong prior $P(\mathbf{v}_n)$ over video candidates as:

$$\log P(\mathbf{v}_n) - \log P(\mathbf{v}_m) > c\varepsilon, \tag{13}$$

where $c > 1$. In other words, the prior gap is larger than the likelihood gap.

Then, the posterior gap, which is the gap of retrieval scores measured by the posterior probability, is given as:

$$\log P(\mathbf{v}_m|\mathbf{t}_m) - \log P(\mathbf{v}_n|\mathbf{t}_m) \tag{14}$$
$$= \log P(\mathbf{t}_m|\mathbf{v}_m) + \log P(\mathbf{v}_m) - \log P(\mathbf{t}_m|\mathbf{v}_n) - \log P(\mathbf{v}_n) \quad \text{(by Eq. (11))} \tag{15}$$
$$< \varepsilon + \log P(\mathbf{v}_m) - \log P(\mathbf{v}_n) \quad \text{(by Eq. (12))} \tag{16}$$
$$< \varepsilon - c\varepsilon = \varepsilon(1 - c) \quad \text{(by Eq. (13))} \tag{17}$$
$$< 0 \quad \text{(by } c > 1) \tag{18}$$

Table 7: **Inference time comparison of Prior Normalized Decoding.** The inference time (seconds per sample) is reported in parentheses. † stands for the model with Prior Normalized Decoding.

| Model | MME | MMBench | MVBench | VideoMME | MLVU | NExT-QA | SeedBench | avg. $\Delta$ |
|---|---|---|---|---|---|---|---|---|
| VideoChat2 (Li et al., 2024b) | 1505.7 (1.5) | 63.9 (1.2) | 60.1 (2.4) | 42.2 (4.1) | 45.8 (6.9) | 78.9 (1.4) | 61.2 (0.9) | - |
| VideoChat2$^\dagger$ (**Ours**) | **1607.0** (2.0) | **66.2** (1.2) | **62.3** (2.4) | **47.1** (4.1) | **48.5** (7.1) | **79.4** (1.5) | **61.7** (1.0) | **+16.3** (+4.9%) |

This concludes the proof. $P(\mathbf{v}_m|\mathbf{t}_m) < P(\mathbf{v}_n|\mathbf{t}_m)$ indicates that the posterior probability $P(\mathbf{v}|\mathbf{t})$ retrieves a wrong $n$-th video $\mathbf{v}_n$ given $m$-th query text $\mathbf{t}_m$ due to the too strong prior $P(\mathbf{v})$. Our Prior Normalization effectively reduces the impact of prior $P(\mathbf{v})$ and improves the retrieval performance.

# G   DICUSSION ON COMPUTATIONAL COST

Since traversing all video candidates requires intensive computational cost, we employ a re-ranking approach to efficiently obtain the retrieval scores among top-$k$ candidates. The VTG and TVG scores generally demonstrate superior performance compared to VTC scores but require a larger computational cost. Therefore, we adopt a hybrid approach which efficiently retrieves the top-$k$ candidates by VTC loss, and then computes more accurate scores using VTG (or TVG) for all pairs of the top-$k$ candidates. This re-ranking approach among top-$k$ candidates significantly reduces the inference time complexity of the single-tower model from $O(mn)$ to $O(k(m+n))$ given $m$ videos and $n$ texts, where $k \ll m, n$. As a result, the inference time is 307 times faster than traversing all candidate videos on ActivityNet (97 hours $\rightarrow$ 19 minutes).

Furthermore, although our final model, BGR$^*$, requires an ensemble of output scores optimized for three different objectives (*i.e.*, VTM, VTG, and TVG), the overall inference process remains efficient due to the shared feature extraction stage across all objectives. In Tab. 3, ensembling VTM, VTG, and TVG boosts R@1 significantly, *e.g.*, 82.4 on DiDeMo, while achieving a 19% improvement in inference speed compared to processing each objective separately.

Finally, Tab. 7 demonstrates the additional inference time overhead of Prior Normalized Decoding on the benchmarks in Tab. 2. Since these benchmarks consist of multi-choice questions, the number of newly generated tokens by the model is less than 10 tokens. This implies that Prior Normalized Decoding introduces only a marginal increase in inference time. In Tab. 7, the average performance is improved by 16.3 while the additional inference time is only increased by 4.9%. On the other hand, the inference time might be increased if the number of newly generated tokens becomes large.

# H   RESULTS ON IMAGE-TEXT RETRIEVAL

Table 8: **Results on image-text retrieval in Flickr30K and COCO.** Prior Normalization is applied to our model variants highlighted in blue cells.

| | Flickr30K | | | | | | COCO | | | | | |
|---|---|---|---|---|---|---|---|---|---|---|---|---|
| | image-to-text | | | text-to-image | | | image-to-text | | | text-to-image | | |
| | R@1 | R@5 | R@10 | R@1 | R@5 | R@10 | R@1 | R@5 | R@10 | R@1 | R@5 | R@10 |
| **generative approach** | | | | | | | | | | | | |
| TIGeR (Qu et al., 2024) | - | - | - | 71.7 | 91.8 | 95.4 | - | - | - | 46.1 | 69.0 | 76.1 |
| GRACE (Li et al., 2024c) | - | - | - | 68.4 | 88.9 | 93.7 | - | - | - | 41.5 | 69.1 | 79.1 |
| **non-generative approach** | | | | | | | | | | | | |
| ALBEF (Li et al., 2021) | 94.1 | 99.5 | 99.7 | 82.8 | 96.3 | 98.1 | 77.6 | 94.3 | 97.2 | 60.7 | 84.3 | 90.5 |
| BLIP (Li et al., 2022) | 96.7 | 100.0 | 100.0 | 86.7 | 97.3 | 98.7 | 82.4 | 95.4 | 97.9 | 65.1 | 86.3 | 91.8 |
| BLIP-2 (Li et al., 2023b) | 97.6 | **100.0** | **100.0** | 89.7 | 98.1 | 98.9 | 85.4 | **97.0** | **98.5** | 68.3 | 87.7 | 92.6 |
| BLIP-2 + Prior Normalization (Ours) | **99.4** | **100.0** | **100.0** | 90.7 | 98.4 | 99.1 | 89.2 | 96.9 | **98.5** | 68.9 | 89.0 | 92.8 |
| **VideoChat2** | | | | | | | | | | | | |
| VTM (Baseline) | 95.9 | **99.7** | **99.7** | 86.3 | 97.2 | 98.4 | 78.5 | 93.6 | 96.6 | 62.0 | 84.7 | 90.7 |
| VTG (**Ours**) | 95.1 | 99.6 | **99.7** | 87.5 | 97.4 | **98.5** | 80.2 | 94.0 | 96.7 | 63.4 | 84.8 | 90.8 |
| TVG (**Ours**) | 97.1 | 99.4 | 99.6 | 80.4 | 95.7 | 97.9 | 79.6 | 93.6 | 96.7 | 56.5 | 82.0 | 89.5 |
| BGR (**Ours**) | 96.8 | 99.6 | **99.7** | 88.8 | 97.6 | **98.5** | 82.6 | 94.6 | 96.8 | 64.6 | 85.5 | 90.9 |
| BGR$^*$ (**Ours**) | **98.7** | **99.7** | **99.7** | **89.3** | **97.8** | **98.5** | **85.1** | **95.1** | **96.9** | **65.9** | **86.1** | **91.3** |

Our BGR with Prior Normalization is a generic framework that can be applied to text-image retrieval in a generative manner. Tab. 8 shows the performance of BGR on text-image retrieval in

Flick3r (Plummer et al., 2015) and COCO (Lin et al., 2014). Our BGR outperforms all the text-image generative retrieval baselines. Furthermore, we compare our model with non-generative text-image retrieval models and our BGR surpasses BLIP-2 both on Flickr30K and COCO in image-to-text retrieval. Interestingly, our Prior Normalization can also be applied to BLIP-2 enhancing performance across all settings and achieving a new state-of-the-art result. These results signify the effectiveness of our BGR and Prior Normalization even in text-image retrieval.

## I Qualitative Results on Instruction-based Retrieval

In this section, we explore the MLLMs' versatility in the human instruction-based retrieval task. We note that the benchmark for human instruction-based retrieval is not yet studied, so we customize ReXTime (Chen et al., 2024a), originally released for the moment-retrieval task, adequately to our setting and we provide qualitative results on several examples. In Fig. 9, we mainly ask the model to retrieve a certain part of the video and the answer given the video and question, *i.e.*, multi-modal queries and multi-modal contents. Specifically, in Fig. 9a, the user asks to retrieve the answer and the relevant part of the video to "What does the man do after walking the tube back?". Our BGR successfully retrieves the relevant part of the video including the 3rd, 4th, and 5th frames along with the text "The man goes up the tow rope.", as the action "walking the tube back" occurs in the 3rd frame. This retrieved video includes the action where the man goes up the tow rope. Furthermore, we ask two different questions with the same video in Fig. 9b and 9c. Our model retrieves the relevant part of the video and the answer well by following the instructions. In Fig. 9b, the scene of gaining momentum for throwing the javelin and the text "To gain momentum for throwing the javelin off into the distance." are retrieved given the question "Why does the person begin running down the track?" and the full video. Interestingly, as the question is changed to "How does the person throw the javelin off into the distance?", the retrieved scene and text are changed to the content depicting "running down the track". Overall, integrating the retrieval task into MLLMs enables them to handle complex human instruction-based retrieval in the real-world chatting system.

## J Negative Impacts and Limitations

**Negative impacts.** Our BGR and Prior Normalization, in themselves, do not have negative societal impacts. However, if there exist biases (*e.g.*, race, gender, religion, and culture) in the benchmark dataset, our model might learn those biases.

**Limitations.** Our final model requires the ensemble of the model's output scores optimized over three different objectives (VTG, TVG, and VTM). Therefore, in inference, scores are obtained from three different objectives, requiring further inference time.

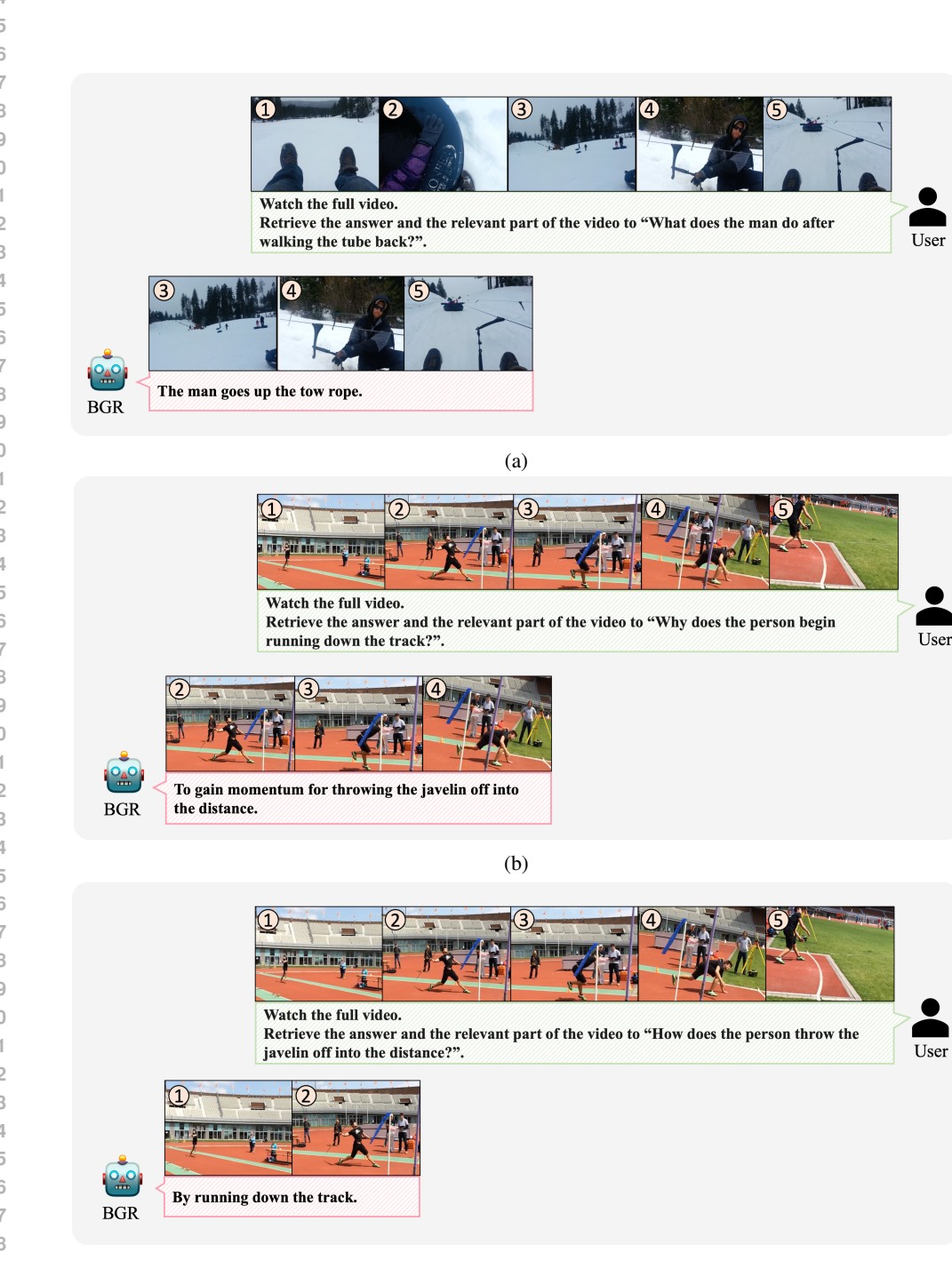

(a)

(b)

(c)

Figure 9: **Qualitative results of human instruction-based retrieval on ReXTime.**

