# OpenReview forum: "Bidirectional Generative Retrieval with Multi-Modal LLMs for Text-Video Retrieval"
_ICLR.cc/2025/Conference — Submitted to ICLR 2025_

### Official Review · Reviewer_ESdv · 2024-10-30

**Soundness:** 3
**Presentation:** 2
**Contribution:** 2
**Rating:** 3
**Confidence:** 5

**Summary:**

We investigated the ability of MLLMs to approach text-video retrieval as a generation task from two perspectives:
1. Content Generation: Generating content directly from a given query；
2. Query Generation: Generating a query from provided content；
We observed that in T2V and V2T tasks, using query generation outperformed content generation, showing better performance and reduced bias. Our framework optimizes from both perspectives: generating VTG (video-to-text generation) for a given video and generating TVG (text-to-video generation) from a given text’s video features. We also integrated a plug-and-play Prior Normalization module to mitigate biases within unimodal data."

**Strengths:**

1. This work provides a comprehensive background on the video-text retrieval task and includes a detailed summary of relevant datasets and evaluation benchmarks for this task in the supplementary materials.

2. The author first applies Prior Normalized Decoding to multimodal scene understanding tasks, offering a solid mathematical foundation and notable improvements in generalization performance.

3. Centered on Prior Normalization, the authors provide extensive ablation studies to substantiate the effectiveness of the module.

**Weaknesses:**

1. The authors' writing logic presents certain issues. I recommend restructuring the entire paper with Prior Normalization as the central focus. From an overall readability perspective, the connection between Prior Normalization and Generative Retrieval appears weak, suggesting that Prior Normalization functions more effectively as a plug-and-play multimodal optimization tool.

2. The authors' definition of the query-generation task is quite confusing. The primary goal of a retrieval task is to retrieve unknown content based on a given query. However, does query-generation entail generating a query in reverse from the content? This approach does not align with the fundamental objective of retrieval tasks.

3. Figure~1 needs modification. Figure 1(a) Text-video retrieval should depict a representation learning model performing matches within the database. Figure 1(c) is also overly confusingly illustrated."

**Questions:**

Why does the simple regularization of Prior Normalization yield performance improvements? The authors need to include relevant references and mathematical analysis on Prior Normalization. I believe this concept is not the authors' original discovery, as it has likely appeared in other fields previously[1].

[1]. Prior normalization for certified likelihood-informed subspace detection of Bayesian inverse problems

---

> ### Author Response · Authors · 2024-11-21
> **Author response to Reviewer ESdv (1/2)**
>
> We appreciate **Reviewer ESdv** for valuable feedback and constructive comments. We will address all the concerns raised by the reviewer, hoping for more vigorous support for our paper.
>
> ---
> 1. **Weak connection between Prior Normalization and Bidirectional Generative Retrieval.**
>
>     For the first time in the context of text-video retrieval with MLLMs, we observe that the model often fails to retrieve the most relevant content given the query due to **prior bias** where the posterior is overwhelmed by the prior of the content. To alleviate the prior bias, we propose both **Bidirectional Generative Retrieval (BGR)** and **Prior Normalization**. Specifically, we apply Prior Normalization with respect to each direction in BGR, *i.e.*, content-generation and query-generation. As a result, alleviating the prior bias in both generative directions leads to outstanding performance across all the text-video retrieval benchmarks.
>
> ---
> 2. **Clarify the query-generation model.**
>
>     The text-video retrieval task assumes the content candidates are given and requires the model to find optimal content among the content candidates. Both the content-generation and query-generation models are used to **find optimal content** among the content candidates. In the **content-generation model**, the optimal content with the maximum likelihood is chosen by replacing the content candidates one by one in the **output** of the model, which is **most likely to be generated by the query**. On the other hand, in the **query-generation model**, the optimal content with the maximum likelihood is chosen by replacing the content candidates one by one in the **input** of the model, which is **most likely to generate the query**. The overall inference procedure is illustrated in Fig. 5 of the supplement. We also have updated Fig. 1 of the main paper and the related description by additionally illustrating content candidates.

---

> > ### Comment · Reviewer_ESdv · 2024-11-22
> > **Response to Author**
> >
> > 1.  Supplementary Figure.5 not Figure.6
> >
> > 2. Your clarification for query-generation retrieval is still confusing. Yet, I have understood the definition of query-generation retrieval based on the discussion with my coworkers. We believe that query-generation retrieval leads to tremendous computational cost for a single text-video retrieval sample due to the addition of text-to-video generation module.
> >
> >
> > The setting of your work is far from the main-stream text-video retrieval. Thus I tend to reject this paper.

---

> ### Author Response · Authors · 2024-11-21
> **Author response to Reviewer ESdv (2/2)**
>
> 3. **Mathematical analysis on Prior Normalization and more references.**
>
>     We focus on searching the most relevant video given a text query, *i.e.*, text-to-video retrieval by $P(\textbf{v}|\textbf{t})$, which can be written as:
>
>     $$
>     \begin{align}
>     P(\mathbf{v}|\mathbf{t})= P(\mathbf{t}|\mathbf{v})P(\mathbf{v})/P(\textbf{t}).\tag{Eq. (1)}
>     \end{align}
>     $$
>
>     As discussed in Tab. 1 of the main paper and Fig. 6 in the supplement, query generation (or likelihood) $P(\textbf{t} | \textbf{v})$  is more accurate than content generation $P(\textbf{v}|\textbf{t})$. In other words, even if $P(\textbf{t}| \textbf{v})$ predicts accurately, the final result $P(\textbf{v}|\textbf{t})$ can lead to incorrect retrievals due to a highly skewed video prior probability $P(\mathbf{v})$, namely, "**prior bias**".
>
>     We will discuss the case where a too large prior gap in $P(\mathbf{v})$ leads to incorrect retrieval results in the posterior $P(\mathbf{v}|\mathbf{t})$ when the likelihood $P(\mathbf{t}|\mathbf{v})$ is correct but the gap is relatively small. Specifically, we first assume that the likelihood predicts correctly but the prior gap is too larger than the likelihood gap. Then we prove that the posterior probability results in a wrong prediction.
>
>     Assume that the likelihood predicts correctly, *i.e.*, $P(\mathbf{t}_m|\mathbf{v}_m) > P(\mathbf{t}_m|\mathbf{v}_n)>0, \forall n \neq m$ but the gap is small. Then, it can be written as:
>
>     $$
>     \begin{align}
>     0 < \log P(\mathbf{t}_m|\mathbf{v}_m) - \log P(\mathbf{t}_m|\mathbf{v}_n) < \varepsilon, \tag{Eq. (2)}
>     \end{align}
>     $$
>
>     where the $\varepsilon > 0$ is the maximum gap. Eq. (2) means the retrieval score of $m$-th video (positive sample) is larger than the one of $n$-th video (negative sample) given the $m$-th text, and the gap is smaller than the maximum gap $\varepsilon$.
>
>     Another assumption is that the strong prior $P(\mathbf{v}_n)$ over video candidates as:
>
>     $$
>     \begin{align}
>     \log P(\mathbf{v}_n) - \log P(\mathbf{v}_m) > c\varepsilon \tag{Eq. (3)}
>     \end{align}
>     $$
>
>     where $c > 1$. In other words, the prior gap is larger than the likelihood gap.
>
>     Then, the posterior gap, which is the gap of retrieval scores measured by the posterior probability, is given as:
>
>     $$
>     \begin{align}
>     &\log P(\mathbf{v}_m|\mathbf{t}_m) - \log P(\mathbf{v}_n|\mathbf{t}_m) &\\\\
>     &\\;= \log P(\mathbf{t}_m|\mathbf{v}_m) + \log P(\mathbf{v}_m) - \log P(\mathbf{t}_m|\mathbf{v}_n) - \log P(\mathbf{v}_n) &\text{(by Eq. (1))} \\\\
>     & \\;\\;< \varepsilon + \log P(\mathbf{v}_m) - \log P(\mathbf{v}_n) & \text{(by Eq. (2))} \\\\
>     & \\;\\; < \varepsilon - c\varepsilon = \varepsilon (1-c) & \text{(by Eq. (3))} \\\\
>     & \\;\\;< 0 &\text{(by c >1)}
>     \end{align}
>     $$
>     This concludes the proof. "$P(\mathbf{v}_m|\mathbf{t}_m) < P(\mathbf{v}_n|\mathbf{t}_m)$" indicates that the posterior probability $P(\mathbf{v}|\mathbf{t})$ retrieves a wrong $n$-th video $\mathbf{v}_n$ given $m$-th query text $\mathbf{t}_m$ due to the too strong prior $P(\mathbf{v})$. Our Prior Normalization effectively reduces the impact of prior $P(\mathbf{v})$ and improves the retrieval performance.
>
>
>     Thank you for suggesting references. This paper is the first work studying prior normalization in the context of text-video retrieval with MLLMs by interpreting the likelihood of a single modality as the prior for calculating the joint probability. As suggested, we will provide a broader view of Prior Normalization that adjusts the strength of the prior distribution including the Bayesian approach [1, 2] or maximum a posteriori (MAP) [3, 4]. For example, [1] proposed an elegant prior normalization technique that transforms non-Gaussian prior into standard Gaussian distributions, enabling LIS-based MCMC sampling for high-dimensional Bayesian inverse problems. We have updated our manuscript by including more references.
>
>     [1] Prior normalization for certified likelihood-informed subspace detection of Bayesian inverse problems
>
>     [2] Normalized Power Prior Bayesian Analysis
>
>     [3] Marginal maximum a posteriori estimation using Markov chain Monte Carlo
>
>     [4] Maximum a posteriori probability estimates in infinite-dimensional Bayesian inverse problem

---

> ### Author Response · Authors · 2024-11-22
> **Response to Reviewer ESdv**
>
> **Clarification on the query-generation model and its computational cost.**
>
>   Thank you for identifying the novelty of our work. We believe that such novelty could be seen as a strength rather than a basis for rejection. Our proposed query-generation model does not lead to tremendous computational costs compared to conventional text-video retrieval models. Conventional text-video retrieval models (*e.g.*, UMT [1], InternVideo [2]) have adopted video-text matching (VTM) loss which estimates joint probability $P(\mathbf{t}, \mathbf{v})$ to measure the relevance score between text and video. On the other hand, in the context of generative retrieval, we propose a content-generation model to estimate the posterior $P(\mathbf{v}|\mathbf{t})$ and a query-generation model to estimate the likelihood $P(\mathbf{t}|\mathbf{v})$ in text-to-video retrieval. The computational cost of the query-generation model is similar to that of the conventional VTM approach, differing only in how the relevance score between text and video is measured. The table below also demonstrates that our query-generation model surpasses the VTM by 2.1 in R@1 despite the comparable inference speed on DiDeMo. We have updated the relevant discussion in Sec. G of the supplement.
>
> | DiDeMo | R@1 | Inference time (sec/sample) |
> | --- | --- | --- |
> | VTM $P(\mathbf{t},\mathbf{v})$ | 74.6 | 127 |
> | Query-generation $P(\mathbf{t} \| \mathbf{v})$  | 76.7 | 120 |
>
> [1] Unmasked teacher: Towards training-efficient video foundation models
>
> [2] InternVideo: Video Foundation Models for Multimodal Understanding

---

### Official Review · Reviewer_Kmgg · 2024-11-02

**Soundness:** 2
**Presentation:** 1
**Contribution:** 2
**Rating:** 3
**Confidence:** 4

**Summary:**

The authors propose a method that addresses the challenge of text-video retrieval using multi-modal large language models. The proposed method leverages the generative capabilities of MLLMs in two directions: content generation (generating the content given a query) and query generation (generating the query given the content). The authors also introduce a simple plug-and-play module called Prior Normalization, which is a training-free score calibration method to further reduce the prior bias at retrieval inference.

**Strengths:**

* The proposed BGR framework, when combined with Prior Normalization, outperforms existing methods on text-video retrieval tasks across various benchmark datasets.
* The idea of using multi-modal large language models for text-video retrieval is interesting.

**Weaknesses:**

* The writing of this paper needs significant improvement:

  * First, this paper defines cross-modal similarity as a conditional probability but does not clearly explain how to calculate it. Is the conditional probability a combination of the LLM's output probabilities for each token, just the probability of the last token, or is it based on the cosine similarity of the LLM output tokens?
  * Second, the paper uses many similar abbreviations, like vtg and tvg, which make it hard to follow.
  * Third, the explanation of the hyper-parameter settings is unclear. For example, $\alpha$ is a key parameter in Prior Normalization that affects the model’s performance, but it’s not clear how its value is chosen. Is $\alpha$ a fixed value, or is it adjusted for each dataset?

* The proposed method isn’t specific to video, so it’s not ideal to limit experiments to only text-video retrieval. Adding text-image retrieval tests would better verify its effectiveness.
* There are some typos in the equations, like missing "( )" in Eq. 1. I strongly recommend checking each equation carefully.
* The proposed method requires traversing all candidate videos to calculate the conditional probability, and Prior Normalization also needs to process each candidate video. This approach is computationally intensive for multi-modal large language models.
* The proposed method uses significantly more parameters and pre-training data than the other methods in Tab. 3. To ensure a fair comparison, I suggest using the same vision encoder (such as BLIP 2 and UMT) as VideoChat2 for the other methods in Tab. 3.

**Questions:**

* How is the value of $\alpha$ chosen?
* What is the inference time for the proposed method on ActivityNet?

---

> ### Author Response · Authors · 2024-11-21
> **Author response to Reviewer Kmgg (1/1)**
>
> We appreciate **Reviewer Kmgg** for valuable feedback and constructive comments. We will address all the concerns raised by the reviewer, hoping for more vigorous support for our paper.
>
> ---
> 1. **Questions on definitions.**
>
>     We provide the definition of the conditional probability $P_\text{vtg}(\mathbf{t}|\mathbf{v})$ in Eq. (3) and $P_\text{tvg}(\mathbf{v}|\mathbf{t})$ in Eq. (5) of the main paper. Specifically, $P_\text{vtg}(\mathbf{t}|\mathbf{v})$ is defined by the product of LLM’s output probabilities and $P_\text{tvg}(\mathbf{v}|\mathbf{t})$ is calculated by the cosine similarity between the LLM output tokens and the video vocabularies. Also, thank you for pointing out the confusing abbreviations and typos. We will update our manuscript with less confusing abbreviations in our final version.
>
> ---
> 2. **Question on the normalization strength $\alpha$.**
>
>     $\alpha$ is a hyperparameter to adjust the normalization strength in our Prior Normalization.  It can be different from each dataset but we observe that $\alpha=1$ generally yields the best performance for all the datasets so we fix $\alpha$ as 1.
>
> ---
> 3. **Add text-image retrieval result.**
>
>     Our BGR with Prior Normalization is a generic framework that can be applied to text-image retrieval in a generative manner. The below table shows the performance of BGR on text-image retrieval (Flick3r and COCO). Our BGR outperforms all the text-image generative retrieval baselines.
>
>     | R@1 | Flickr30K (V2T) | COCO (V2T) | Flickr30K (T2V) | COCO (T2V) |
>     | --- | --- | --- | --- | --- |
>     | GRACE  | - | - | 68.4 | 41.5 |
>     | TIGeR | - | - | 71.7 | 46.1 |
>     | VTM (**ours**) | 95.9 | 78.5 | 86.3 | 62.0 |
>     | VTG (**ours**) | 95.1 | 80.2 | 87.5 | 63.4 |
>     | TVG (**ours**) | 97.1 | 79.6 | 80.4 | 56.5 |
>     | BGR (**ours**) | 96.8 | 82.6 | 88.8 | 64.6 |
>     | BGR$^*$ (**ours**) | **98.7** | **85.1** | **89.3** | **65.9** |
>
>     Furthermore, we compare our model with non-generative text-image retrieval models in the below table and our BGR surpasses BLIP-2 both on Flickr30K and COCO in V2T. Interestingly, our Prior Normalization can also be applied to BLIP-2 enhancing performance across all settings and achieving a new state-of-the-art result.
>
>     | R@1 | Flickr30K (V2T) | COCO (V2T) | Flickr30K (T2V) | COCO (T2V) |
>     | --- | --- | --- | --- | --- |
>     | ALBEF | 94.1 | 77.6 | 82.8 | 60.7 |
>     | BLIP | 96.7 | 82.4 | 86.7 | 65.1 |
>     | BLIP-2 | 96.9 | 83.5 | 89.7 | 68.3 |
>     | BLIP-2 + Prior Normalization (**ours**) | **99.4** | **89.2** | **90.7** | **68.9** |
>     | VTM (**ours**) | 95.9 | 78.5 | 86.3 | 62.0 |
>     | VTG (**ours**) | 95.1 | 80.2 | 87.5 | 63.4 |
>     | TVG (**ours**) | 97.1 | 79.6 | 80.4 | 56.5 |
>     | BGR (**ours**) | 96.8 | 82.6 | 88.8 | 64.6 |
>     | BGR$^*$ (**ours**) | **98.7** | **85.1** | **89.3** | **65.9** |
>
>     These results signify the effectiveness of our BGR and Prior Normalization even in text-image retrieval.
>
> ---
> 4. **Discuss computational cost.**
>
>     Since traversing all video candidates requires intensive computational cost, we employ a re-ranking approach to efficiently obtain the retrieval scores among top-$k$ candidates. Specifically, as discussed in Sec. 2.1 of the main paper, we adopt a hybrid approach which efficiently retrieves the top-$k$ candidates by VTC loss, and then computes more accurate scores using VTG (or TVG) for all pairs of the top-$k$ candidates. This re-ranking approach among top-$k$ candidates significantly reduces the inference time complexity of the single-tower model from $O(mn)$ to $O(k(m+n))$ given $m$ videos and $n$ texts, where $k \ll m, n$. As a result, the inference time is 307 times faster than traversing all candidate videos on ActivityNet (97 hours → 19 minutes).
>
> ----
> 5. **Performance comparison with the same vision encoder.**
>
>     In Tab. 3 of the main paper, the baseline MLLM, referred to as VTM, shares the **exact** same architecture (including the vision encoder) as our BGR, *i.e.*, UMT + Q-Former + LLM, but is trained with a conventional retrieval approach, video-text matching loss described in Sec. 3.1. While the average R@1 of the VTM baseline is lower than the one of UMT by 2.2, our BGR with Prior Normalization outperforms VTM by 6.6 in R@1 on average. This implies that the performance improvement mainly comes from our BGR and Prior Normalization framework not the model architecture with the strong vision encoder.

---

> > ### Comment · Reviewer_Kmgg · 2024-11-23
> >
> > I appreciate the authors' responses, but I still have some concerns.
> >
> > First, I remain concerned about the model's inference efficiency. As noted in lines 324 to 326 of the paper, the Prior Normalization requires traversing the entire collection of text and video during retrieval. The re-ranking approach only reduces the number of candidate texts, but it does not address the number of query videos, which limits its efficiency.
> >
> > Second, I have a new concern regarding the use of a set of queries when calculating normalization coefficients. In my view, it is unreasonable to use information from other queries to retrieve candidate samples for a single query. Moreover, in the retrieval field, group queries are generally considered superior to single queries, and there has been significant recent work on normalization techniques for group queries, such as QB-Norm [1] and DSL [2].
> >
> > When comparing model performance, we usually exclude group query normalization for a fairer evaluation. This is because group queries can artificially leverage dataset vulnerabilities. Specifically, in the dataset used in this paper, video and text are one-to-one correspondences, and using group query normalization effectively amounts to conducting retrieval in two directions, exploiting this one-to-one correspondence. The absence of experiments on one-to-many retrieval datasets, such as MSVD, further raises my concerns.
> >
> > [1] Simion-Vlad Bogolin, Ioana Croitoru, Hailin Jin, Yang Liu, and Samuel Albanie. Cross modal retrieval with querybank normalisation. In CVPR, pages 5194–5205, 2022.
> >
> > [2] Xing Cheng, Hezheng Lin, Xiangyu Wu, Fan Yang, and Dong Shen. Improving videotext retrieval by multi-stream corpus alignment and dual softmax loss. arXiv preprint arXiv:2109.04290, 2021.

---

> ### Author Response · Authors · 2024-11-25
> **Author response to Reviewer Kmgg**
>
> **Discussion on Prior Normalization and group queries.**
>
>   Similar to the re-ranking approach that uses only $k$ candidates for traversing, we also sample $k$ queries to calculate coefficients for Prior Normalization which significantly reduces the computational cost compared to considering all the queries. We do understand that group query methods are superior to single queries, but we highlight that our model surpasses all the baselines even without Prior Normalization (*i.e.*, single query setting), and the performance is further improved by applying it. The table below demonstrates the performance of UMT and BGR$^*$ w/o Prior Normalization on five retrieval datasets including MSVD, one-to-many retrieval setting, for a fair comparison. Our BGR$^*$ significantly outperforms UMT by a margin of 5.3 in R@1 on average across all the datasets, achieving a new state-of-the-art performance.
>
> | T2V R@1 | avg. R@1 | DiDeMo | ActivityNet | LSMDC | MSRVTT | MSVD |
> | --- | --- | --- | --- | --- | --- | --- |
> | UMT w/o Prior Normalization | 59.2 | 70.6 | 66.8 | 43.0 | 58.8 | 56.6 |
> | BGR$^*$ w/o Prior Normalization | **64.5** | **79.2** | **73.2** | **49.1** | **60.6** | **60.5** |
>
> Furthermore, by applying Prior Normalization to both BGR$^*$ and UMT, the performances are increased in both models and our BGR$^*$ still surpasses UMT on average, as discussed in Tab. 5 and 6 of the supplement. The table below summarizes the performance of BGR$^*$ with Prior Normalization as well as other group query methods QB-Norm and DSL. The table shows our BGR$^*$ w/ Prior Normalization outperforms UMT w/ Prior Normalization, while showing superior performance compared to other group query methods.
>
> | T2V R@1 | avg. R@1 | DiDeMo | ActivityNet | LSMDC | MSRVTT | MSVD |
> | --- | --- | --- | --- | --- | --- | --- |
> | UMT w/ Prior Normalization | 62.7 | 75.4 | 73.2 | 45.1 | 62.4 | 57.3 |
> | BGR$^*$ w/ QB-Norm | 65.0 | 81.4 | 75.4 | 49.3 | 60.3 | 58.7 |
> | BGR$^*$ w/ DSL | 65.3 | 80.7 | 72.2 | 50.8 | **62.3** | 60.5 |
> | BGR$^*$ w/ Prior Normalization | **66.4** | **82.4** | **75.9** | **51.1** | 62.0 | **60.7** |

---

> > ### Comment · Reviewer_Kmgg · 2024-11-27
> >
> > I appreciate the authors' responses.
> >
> > The proposed method is built upon VideoChat2, with UMT as its vision encoder. The visual features encoded by UMT are then processed by VideoChat2 through a large language model (LLM). Given that UMT is the vision encoder of the proposed method, it is reasonable to predict that the proposed method would outperform UMT.
> > I have another three questions：
> >
> > * First, I noticed a significant discrepancy between the MSVD performance comparison provided by the authors and the results presented in the UMT paper. In the UMT paper, UMT achieves a performance of 80.3 instead of 56.6 on MSVD.
> > * Second, in my experiments, the performance of QB-Norm was significantly lower than DSL, can you share the hyper-parameter settings of the QB-Norm you used?
> > * Third, I still believe that the use of group queries is unreasonable, as it assumes that queries are independent of each other. In the MSVD dataset, each video in the test set corresponds to multiple query texts. How did you address the issue of group query failure caused by the excessive similarity between query texts?

---

> ### Author Response · Authors · 2024-11-28
> **Author response to Reviewer Kmgg**
>
> 1. **Performance of UMT on MSVD. (Q1)**
>
>     We observe that the UMT paper was updated on March 11, 2024, due to the bug reported in its GitHub, and the performance on MSVD is 58.2 in T2V. 56.6 in our table is reproduced performance by us. Please refer to UMT [GitHub](https://github.com/OpenGVLab/unmasked_teacher?tab=readme-ov-file) and Tab. 14 of the [paper](https://arxiv.org/pdf/2303.16058) for more details.
>
> 2. **Hyper-parameter settings and the issue of group query failure. (Q2 and Q3)**
>
>     Our Prior Normalization does not assume the one-to-one correspondence of the dataset. Unlike previous approaches like QB-Norm and DSL, which explicitly enforce one-to-one correspondence through group queries, we use other queries to calibrate the posterior probability, preventing the posterior probability from being overwhelmed by the prior probability. Therefore, our Prior Normalization successfully handles the issue of group query failure caused by the many-to-one correspondences in MSVD. Furthermore, although we have used the default settings of QB-Norm and DSL without using temperature in the Softmax operation, we observe that our BGR is robust to such group query methods by considering the posterior and likelihood at the same time in a bidirectional manner, already alleviating the overwhelming problem of the prior probability to some extent.

---

### Official Review · Reviewer_xKPZ · 2024-11-03

**Soundness:** 3
**Presentation:** 2
**Contribution:** 2
**Rating:** 5
**Confidence:** 3

**Summary:**

The paper introduces the Bidirectional Text-Video Generative Retrieval (BGR) framework to enhance text-video retrieval tasks by mitigating prior biases in MLLMs through a dual-generation approach (VTG and TVG) and a plug-and-play Prior Normalization module. Extensive experiments on benchmark datasets demonstrate that BGR significantly outperforms traditional methods, showcasing improved performance in text-to-video and video-to-text retrieval.

**Strengths:**

1. Authors provide a baseline model using MLLM.
2. Authors provide solutions for generative retrieval by text video feature generation and video text generation.
3. Authors present a solution for reducing the bias in MLLM.

**Weaknesses:**

1. Lack of discussion and comparison on image-text generative retrieval. For example, comparing with them showing some special design for video (as this is the only difference between video/image text generative retrieval.)
3. Writing is not clear.
	1. Lack of explanation of lots of mathematical  formulations. For example, what is v_1, v_2, t_1, t_2 in Figure 3? What is the difference between them and tilde{t}^1 and tilde{t}^2 . The introduction of these mathematical notations should be before Figure 2.
	2. In figure 3, how to get t_2, t_3, v_2, v_3 from tilde t _ 1. v_1?
	3. The definition of P(t) in Line 187 is not clear. How to calculate that?
	4. The details of several models should be put in the main paper not the appendix.
.

**Questions:**

First see weakness.
Q1: So in 3.1, the MMLM baseline is a single tower model? The complexity during training and inferences is O(mn) assuming m is the number of video and n is number of text? This is way higher that dual tower model.
Q2: How to calculate P(t) in Eq. 10?

---

> ### Author Response · Authors · 2024-11-21
> **Author response to Reviewer xKPZ (1/1)**
>
> We appreciate **Reviewer xKPZ** for valuable feedback and constructive comments. We will address all the concerns raised by the reviewer, hoping for more vigorous support for our paper.
>
> ---
> 1. **Compare with text-image generative retrieval.**
>
>     Our BGR with Prior Normalization is a generic framework that can be applied to text-image retrieval in a generative manner. The below table shows the performance of BGR on text-image retrieval (Flick3r and COCO). Our BGR outperforms all the text-image generative retrieval baselines.
>
>     | R@1 | Flickr30K (V2T) | COCO (V2T) | Flickr30K (T2V) | COCO (T2V) |
>     | --- | --- | --- | --- | --- |
>     | GRACE  | - | - | 68.4 | 41.5 |
>     | TIGeR | - | - | 71.7 | 46.1 |
>     | VTM (**ours**) | 95.9 | 78.5 | 86.3 | 62.0 |
>     | VTG (**ours**) | 95.1 | 80.2 | 87.5 | 63.4 |
>     | TVG (**ours**) | 97.1 | 79.6 | 80.4 | 56.5 |
>     | BGR (**ours**) | 96.8 | 82.6 | 88.8 | 64.6 |
>     | BGR$^*$ (**ours**) | **98.7** | **85.1** | **89.3** | **65.9** |
>
>     Furthermore, we compare our model with non-generative text-image retrieval models in the below table and our BGR surpasses BLIP-2 both on Flickr30K and COCO in V2T. Interestingly, our Prior Normalization can also be applied to BLIP-2 enhancing performance across all settings and achieving a new state-of-the-art result.
>     | R@1 | Flickr30K (V2T) | COCO (V2T) | Flickr30K (T2V) | COCO (T2V) |
>     | --- | --- | --- | --- | --- |
>     | ALBEF | 94.1 | 77.6 | 82.8 | 60.7 |
>     | BLIP | 96.7 | 82.4 | 86.7 | 65.1 |
>     | BLIP-2 | 96.9 | 83.5 | 89.7 | 68.3 |
>     | BLIP-2 + Prior Normalization (**ours**) | **99.4** | **89.2** | **90.7** | **68.9** |
>     | VTM (**ours**) | 95.9 | 78.5 | 86.3 | 62.0 |
>     | VTG (**ours**) | 95.1 | 80.2 | 87.5 | 63.4 |
>     | TVG (**ours**) | 97.1 | 79.6 | 80.4 | 56.5 |
>     | BGR (**ours**) | 96.8 | 82.6 | 88.8 | 64.6 |
>     | BGR$^*$ (**ours**) | **98.7** | **85.1** | **89.3** | **65.9** |
>
>     These results signify the effectiveness of our BGR and Prior Normalization even in text-image retrieval.
>
> ---
> 2. **Questions on definitions.**
>
>     We defined $v_1$, $v_2$, $t_1$, and $t_2$ in Line 221 of Sec. 3.1 of the main paper, which denote the features of the input video and text tokens. Furthermore, $\tilde{t}_1$ and $\tilde{t}_2$ are the output features of the LLM from $t_1$ and $t_2$, respectively, which are defined in Line 243. In Fig. 3, due to the auto-regressive property of the LLM, the output feature of the LLM predicts the next token, *i.e.*, $\tilde{t}_1$ predicts $t_2$, and similarly in $\tilde{v}_1$. Also, we interpret the likelihood of a single-modality as the **prior probability** for calculating the joint probability of a multi-modality. The definition of the prior probability $P(\mathbf{t})$ is provided in Line 310 and 321 of Sec. 3.3.
>
>     Specifically, in text-video retrieval, we marginalize the text-video correspondences $P(\mathbf{v}, \mathbf{t})$ across $\mathbf{v}$ to obtain $P(\mathbf{t})$, *i.e.*, $P(\mathbf{t}) \triangleq \sum_\mathbf{v} P(\mathbf{v}, \mathbf{t})$
>
>
> ---
> 3. **Discuss time complexity between single-tower and dual-tower models.**
>
>     Since the single-tower model generally requires further inference time than the dual-tower model, we employ a re-ranking approach to efficiently obtain the retrieval scores among top-$k$ candidates. Specifically, as discussed in Sec. 2.1 of the main paper, we adopt a hybrid approach which first retrieves the top-$k$ candidates by VTC loss from the dual-tower model, and then computes more accurate scores using VTG (or TVG) from the single-tower model for all pairs of the top-$k$ candidates. This re-ranking approach among top-$k$ candidates significantly reduces the inference time complexity of the single-tower model from $O(mn)$ to $O(k(m+n))$ given $m$ videos and $n$ texts, where $k \ll m, n$. The below table shows the performance comparison between the single-tower approach with the re-ranking strategy and the dual-tower approach of our BGR. With the efficient re-ranking strategy, the R@1 is remarkably increased by 22.9 on ActivityNet.
>     | ActivityNet | training | inference | R@1 |
>     | --- | --- | --- | --- |
>     | dual-tower | $O(m + n)$ | $O(m + n)$ | 53.0 |
>     | single-tower | $O(m + n)$ | $O(k(m+n))$ | 75.9 |

---

> > ### Comment · Reviewer_xKPZ · 2024-11-27
> >
> > Thank the authors for the response. About the time complexity, what is the base model used for retrieval if the proposed method is for reranking?

---

> > > ### Author Response · Authors · 2024-11-27
> > > **Author response to Reviewer xKPZ**
> > >
> > > **The base model for retrieval.**
> > >
> > > Our BGR can serve as both single-tower and dual-tower models, performing both retrieval and re-ranking in one model. The VTC loss introduced in Sec. 2.1 is regarded as a dual-tower model since there is no cross-modal information fusion, and the VTG and TVG losses are regarded as a single-tower model which generates the other modality with a cross-modal information fusion.

---

### Official Review · Reviewer_XdBX · 2024-11-09

**Soundness:** 3
**Presentation:** 3
**Contribution:** 3
**Rating:** 8
**Confidence:** 3

**Summary:**

This paper addresses text-video retrieval using Multi-Modal Large Language Models (MLLMs), focusing on mitigating prior biases such as text bias in video-to-text (V2T) and video bias in text-to-video (T2V). The key contribution is a Bidirectional Generative Retrieval (BGR) framework that integrates content generation (e.g., generating text from video) and query generation (e.g., generating video features from text) within a unified training framework. This bidirectional approach inherently addresses modality-specific biases by leveraging complementary strengths of both directions.

The authors introduce Prior Normalization as a core component to further balance contributions from text and video, enhancing robustness and retrieval accuracy. The framework achieves state-of-the-art results on standard benchmarks (e.g., DiDeMo, ActivityNet, MSRVTT), significantly outperforming contrastive and hybrid methods. By combining bidirectional generation with effective bias mitigation, the paper presents a novel and impactful paradigm for multi-modal retrieval.

**Strengths:**

1. **Innovative Framework**: The integration of content and query generation into a single bidirectional retrieval framework is novel and effective.
2. **Bias Mitigation**: The use of Prior Normalization directly addresses modality-specific biases, improving robustness and accuracy.
3. **State-of-the-Art Results**: The method achieves strong performance across multiple benchmarks, demonstrating its practical effectiveness.

**Weaknesses:**

1. **Low Coverage in V2T**: Generated text queries may oversimplify video content, reducing detail and diversity in retrieval results.
2. **Superficial Signals in T2V**: The approach may overfit to visually striking but semantically irrelevant features in videos.
3. **Limited Discussion of Trade-Offs**: The paper does not analyze the computational complexity or scalability of the bidirectional inference framework. Not critical but should cover.
4. **Omitted Baseline Comparisons**: The evaluation misses DiffusionRet (ICCV 2023) and GLSCL (CVPR 2023), which are relevant recent works but likely not critical given the strong baselines included.

**Questions:**

I generally liked the idea of combining bidirectional generative retrieval with prior normalization—it reminds me of cycle training, though your method doesn’t fully address the problem of ensuring mutual semantic consistency across directions.

I'd like the authors to address a more philosophical question: the information content in videos and queries are vastly different, in terms of diversity, richness etc. i.e. a picture is worth a thousand words and videos (saving for academic talks) will have much more. It wont' be a good idea if we cannot identify "man walking" if that is indeed what the user is asking, instead of some specific aspects. This asymmetric problem exists in both direction, but with different aspects. Trained in this framework, we might "overfit" to the queries in the training data.

---

> ### Author Response · Authors · 2024-11-21
> **Author response to Reviewer XdBX (1/1)**
>
> We appreciate **Reviewer XdBX** for valuable feedback and constructive comments. We will address all the concerns raised by the reviewer, hoping for more vigorous support for our paper.
>
> ---
> 1. **Discuss the asymmetric information problem between text and video modalities.**
>
>     As noted by **Reviewer XdBX**, we also observe that some captions are overly simple and relatively ambiguous compared to videos. These captions often exhibit high prior probabilities and are easily retrieved by most videos. However, the **query-generation model** mitigates this issue by alleviating prior bias and encouraging one-to-one correspondence in retrieval results. We have updated **Fig. 6a in Sec. E of the supplement** visualizing that one caption is predominantly retrieved by most videos with the content-generation model. On the other hand, in **Fig. 6b of the supplement**, adopting the query-generation model alleviates this issue resulting in the optimal retrieval result where each caption is retrieved by its own paired video despite the imbalanced quantity of information in each modality. Exploring training frameworks to address the issue of asymmetric information will be an intriguing direction for future research.
>
> ---
> 2. **Does the proposed approach exhibit overfitting to visually striking videos?**
>
>     In fact, the baseline MLLM easily retrieves the visually **“common”** video (not visually striking) due to the video prior bias problem, and our goal is to prevent the model from such overfitting (bias towards common videos). Interestingly, the video prior bias is relatively less pronounced compared to the text prior bias as shown in Fig. 6a on V2T and 6c on T2V of the supplement. A distinct vertical line is shown in Fig. 6a, absent in Fig. 6c, implying that the baseline MLLM exhibits greater overfitting to text than video.
>
> ---
> 3. **Discuss performance-time trade-off.**
>
>     Although our final model, BGR$^*$, requires an ensemble of output scores optimized for three different objectives, the overall inference process remains efficient due to the shared feature extraction stage across all objectives. In the below table, ensembling VTM, VTG, and TVG boosts R@1 significantly to 82.4, while achieving a 19% improvement in inference speed compared to processing each objective separately.
>
>     | DiDeMo | R@1 | Inference time (sec/sample) |
>     | --- | --- | --- |
>     | VTM | 74.6 | 127 |
>     | VTG | 76.7 | 120 |
>     | TVG | 56.5 | 78 |
>     | BGR (VTG + TVG) | 78.6 | 167 |
>     | BGR$^*$ (VTM + VTG + TVG) | 82.4 | 262 |
>
> ---
> 4. **Compare with relevant works.**
>
>     GLSCL proposed a simple yet effective parameter-free global interaction module for efficient text-video matching and DiffusionRet introduced a novel approach by formulating the retrieval task as a process of gradually generating joint distribution from noise. The below table shows that our BGR$^*$ significantly outperforms both GLSCL and DiffusionRet across all the datasets.
>
>     | R@1 on T2V | DiDeMo | ActivityNet | LSMDC | MSRVTT |
>     | --- | --- | --- | --- | --- |
>     | GLSCL | 46.7 | 41.4 | 23.4 | 50.5 |
>     | DiffusionRet | 48.9 | 48.1 | 24.4 | 49.0 |
>     | UMT | 70.4 | 66.8 | 43.0 | 58.8 |
>     | BGR$^*$ (**ours**) | **82.4** | **75.9** | **51.1** | **62.0** |

---

### Official Review · Reviewer_hsa3 · 2024-11-11

**Soundness:** 1
**Presentation:** 2
**Contribution:** 1
**Rating:** 5
**Confidence:** 4

**Summary:**

This paper proposes a framework, Bidirectional Text-Video Generative Retrieval (BGR), to address the challenge of text-video retrieval using MLLMs. BGR leverages two generation directions: text-to-video (VTG) and video-to-text (TVG). By combining the strengths of both directions and employing a Prior Normalization technique, BGR mitigates the bias towards uni-modal content and improves retrieval performance.

**Strengths:**

1. The proposed method is straightforward and intuitive.
2. The paper is easy to follow.
3. Some empirical results are strong.

**Weaknesses:**

1. The overall contribution is rather limited. The key idea of adding content generation loss is implemented as contrastive losses, which are not fundamentally different from widely-used formulations in video language modeling. The reviewer would expect further exploration on how content generation is uniquely modeled if the main claim is on formulating retrieval as a generative task.

2. More in-depth analysis are needed in the prior normalization technique. It is still unclear why this is needed mathematically. From the current manuscript, it is not well analyzed and explained why it is necessary.

3. It is important to understand how the proposed decoding process affects the inference speed.

**Questions:**

Please check weakness for details.

---

> ### Author Response · Authors · 2024-11-21
> **Author response to Reviewer hsa3 (1/2)**
>
> We appreciate **Reviewer hsa3** for valuable feedback and constructive comments. We will address all the concerns raised by the reviewer, hoping for more vigorous support for our paper.
>
> ---
> 1. **Discuss the novelty of the proposed method.**
>
>     To the best of our knowledge, this is the first work that discusses the **prior bias** in text-video retrieval by MLLMs. We observe that the model often fails to retrieve the most relevant content given the query due to **prior bias** where the posterior is overwhelmed by the strong prior of the content. To alleviate the prior bias problem, we propose several remedies. First, we investigate that the **query-generation model** is less biased toward such prior bias and exhibits more powerful performance than the **content-generation model**. Also, we propose **Prior Normalization**, a simple yet effective plug-and-play module, to further alleviate the prior bias problem even in the content-generation model. Finally, by integrating the query-generation and content-generation models along with Prior Normalization, we present **Bidirectional Generative Retrieval** which demonstrates outstanding performance across all the text-video retrieval benchmarks.
> ---
> 2. **Provide mathematical analysis on Prior Normalization.**
>
>     We focus on searching the most relevant video given a text query, *i.e.*, text-to-video retrieval by $P(\textbf{v}|\textbf{t})$, which can be written as:
>
>     $$
>     \begin{align}
>     P(\mathbf{v}|\mathbf{t})= P(\mathbf{t}|\mathbf{v})P(\mathbf{v})/P(\textbf{t}).\tag{Eq. (1)}
>     \end{align}
>     $$
>
>     As discussed in Tab. 1 of the main paper and Fig. 6 in the supplement, query generation (or likelihood) $P(\textbf{t} | \textbf{v})$  is more accurate than content generation $P(\textbf{v}|\textbf{t})$. In other words, even if $P(\textbf{t}| \textbf{v})$ predicts accurately, the final result $P(\textbf{v}|\textbf{t})$ can lead to incorrect retrievals due to a highly skewed video prior probability $P(\mathbf{v})$, namely, "**prior bias**".
>
>     We will discuss the case where a too large prior gap in $P(\mathbf{v})$ leads to incorrect retrieval results in the posterior $P(\mathbf{v}|\mathbf{t})$ when the likelihood $P(\mathbf{t}|\mathbf{v})$ is correct but the gap is relatively small. Specifically, we first assume that the likelihood predicts correctly but the prior gap is too larger than the likelihood gap. Then we prove that the posterior probability results in a wrong prediction.
>
>     Assume that the likelihood predicts correctly, *i.e.*, $P(\mathbf{t}_m|\mathbf{v}_m) > P(\mathbf{t}_m|\mathbf{v}_n)>0, \forall n \neq m$ but the gap is small. Then, it can be written as:
>
>     $$
>     \begin{align}
>     0 < \log P(\mathbf{t}_m|\mathbf{v}_m) - \log P(\mathbf{t}_m|\mathbf{v}_n) < \varepsilon, \tag{Eq. (2)}
>     \end{align}
>     $$
>
>     where the $\varepsilon > 0$ is the maximum gap. Eq. (2) means the retrieval score of $m$-th video (positive sample) is larger than the one of $n$-th video (negative sample) given the $m$-th text, and the gap is smaller than the maximum gap $\varepsilon$.
>
>     Another assumption is that the strong prior $P(\mathbf{v}_n)$ over video candidates as:
>
>     $$
>     \begin{align}
>     \log P(\mathbf{v}_n) - \log P(\mathbf{v}_m) > c\varepsilon \tag{Eq. (3)}
>     \end{align}
>     $$
>
>     where $c > 1$. In other words, the prior gap is larger than the likelihood gap.
>
>     Then, the posterior gap, which is the gap of retrieval scores measured by the posterior probability, is given as:
>
>     $$
>     \begin{align}
>     &\log P(\mathbf{v}_m|\mathbf{t}_m) - \log P(\mathbf{v}_n|\mathbf{t}_m) &\\\\
>     &\\;= \log P(\mathbf{t}_m|\mathbf{v}_m) + \log P(\mathbf{v}_m) - \log P(\mathbf{t}_m|\mathbf{v}_n) - \log P(\mathbf{v}_n) &\text{(by Eq. (1))} \\\\
>     & \\;\\;< \varepsilon + \log P(\mathbf{v}_m) - \log P(\mathbf{v}_n) & \text{(by Eq. (2))} \\\\
>     & \\;\\; < \varepsilon - c\varepsilon = \varepsilon (1-c) & \text{(by Eq. (3))} \\\\
>     & \\;\\;< 0 &\text{(by c >1)}
>     \end{align}
>     $$
>
>     This concludes the proof. "$P(\mathbf{v}_m|\mathbf{t}_m) < P(\mathbf{v}_n|\mathbf{t}_m)$" indicates that the posterior probability $P(\mathbf{v}|\mathbf{t})$ retrieves a wrong $n$-th video $\mathbf{v}_n$ given $m$-th query text $\mathbf{t}_m$ due to the too strong prior $P(\mathbf{v})$. Our Prior Normalization effectively reduces the impact of prior $P(\mathbf{v})$ and improves the retrieval performance.

---

> ### Author Response · Authors · 2024-11-21
> **Author response to Reviewer hsa3 (2/2)**
>
> 3. **Analyze the inference time of Prior Normalized Decoding.**
>
>     Since the number of newly generated tokens is less than 10 in our experiment in Tab. 2, the additional overhead in inference time is marginal. Specifically, all the benchmarks consist of multi-choice questions, requiring the model to generate a short sequence. This results in only 4.9% overhead in the inference time while achieving a 16.3 improvement in the performance on average. Note that a greater number of token generations may lead to an increase in inference time.
>
>     |  | avg. $\Delta$ | MME | MMBench | MVBench | VideoMME | MLVU | NExT-QA | SeedBench |
>     | --- | --- | --- | --- | --- | --- | --- | --- | --- |
>     | VideoChat2 w/o Prior Normalized Decoding | - | 1505.7 (1.5 sec/sample) | 63.9 (1.2 sec/sample) | 60.1 (2.4 sec/sample) | 42.2 (4.1 sec/sample) | 45.8 (6.9 sec/sample) | 78.9 (1.4 sec/sample) | 61.2 (0.9 sec/sample) |
>     | VideoChat2 w/ Prior Normalized Decoding | +16.3 (+4.9%) | 1607.0 (2.0 sec/sample) | 66.2 (1.2 sec/sample) | 62.3 (2.4 sec/sample) | 47.1 (4.1 sec/sample) | 48.5 (7.1 sec/sample) | 79.4 (1.5 sec/sample) | 61.7 (1.0 sec/sample) |

---

### Author Response · Authors · 2024-11-21
**Global response by authors**

We sincerely appreciate the reviewers for their time and efforts in reviewing our paper with constructive comments. We have responded to all the questions and concerns raised by reviewers and uploaded the revised version of our manuscript (colored in blue for better visibility). The major changes in our manuscript are summarized below:

1. Improve the writing of the paper.
    - We have updated Sec. 1 and Fig. 1 to explain that the model retrieves the content among content candidates (**Reviewer ESdv**).
    - We have updated Sec. 3.2 to describe details of the inference procedure of the query-generation model (**Reviewer ESdv, xKPZ**).
    - We have updated some equations (*e.g.*, Eq. (1)) by fixing typos (**Reviewer Kmgg**).
2. Improve the analysis of the prior bias problem.
    - We have updated Sec. 1 and Fig. 2 / Sec. E and Fig. 6 to support the explanation of the prior bias problem (**Reviewer XdBX)**.
3. Include a discussion on Prior Normalization by comparing it with relevant studies.
    - We have updated Sec. 3.3 to compare our Prior Normalization with relevant studies (**Reviewer** **ESdv**).
4. Include a discussion on computational cost.
    - We have updated Sec. G to provide a discussion on the time complexity of the re-ranking approach (**Reviewer** **xKPZ, Kmgg**).
    - We have updated Sec. G to include an analysis of the performance-time trade-off by retrieval score ensembling (**Reviewer** **XdBX**).
    - We have updated Sec. G to analyze the inference time of Prior Normalized Decoding (**Reviewer** **hsa3**).
5. Apply our framework to text-image retrieval.
    - We have updated Sec. H to provide an additional experiment on text-image retrieval (**Reviewer** **xKPZ, Kmgg)**.
6. Add mathematical analysis on Prior Normalization.
    - We have updated Sec. F of the supplement for mathematical analysis on Prior Normalization (**Reviewer hsa3, ESdv**).

We believe that incorporating this constructive feedback significantly enhances the quality of our paper. Please find individual responses to the comments below.

---

### Meta-Review · Area_Chair_gbY4 · 2024-12-19

**Metareview:**

This work aims to leverage MLLMs for text-video retrieval with the proposed Bidirectional Generative Retrieval (BGR) framework. While Reviewer XdBX liked the main idea, reviewers shared some concerns, e.g., more in-depth analysis, improving writing, inference efficiency, etc. After rebuttal, those concerns are not fully addressed. AC agrees that this work is not ready for publish and suggests authors to improve the work according to suggestions from reviewers.

**Additional Comments On Reviewer Discussion:**

Reviewer xKPZ, Kmgg and ESdv discussed with authors during rebuttal on efficiency, performance and problem definition. However, those reviewers still leaned to reject the work after discussion.

---

### Decision · Program_Chairs · 2025-01-22

Reject